# GENERALIST EQUIVARIANT TRANSFORMER TOWARDS 3D MOLECULAR INTERACTION LEARNING

## ABSTRACT

Many processes in biology and drug discovery involve various 3D interactions between molecules, such as protein and protein, protein and small molecule, etc. Given that different molecules are usually represented in different granularity, existing methods usually encode each type of molecules independently with different models, leaving it defective to learn the universal underlying interaction physics. In this paper, we first propose to universally represent an arbitrary 3D complex as a geometric graph of sets, shedding light on encoding all types of molecules with one model. We then propose a Generalist Equivariant Transformer (GET) to effectively capture both domain-specific hierarchies and domain-agnostic interaction physics. To be specific, GET consists of a bilevel attention module, a feed-forward module and a layer normalization module, where each module is E(3) equivariant and specialized for handling sets of variable sizes. Notably, in contrast to conventional pooling-based hierarchical models, our GET is able to retain fine-grained information of all levels. Extensive experiments on the interactions between proteins, small molecules and RNA/DNAs verify the effectiveness and generalization capability of our proposed method across different domains.

## 1 INTRODUCTION

Molecular interactions Tomasi & Persico (1994), which describe attractive or repulsive forces between molecules and between non-bonded atoms, are crucial in the research of chemistry, biochemistry and biophysics, and come as foundation processes of various downstream applications, including drug discovery, material design, etc (Sapoval et al., 2022; Tran et al., 2023; Vamathevan et al., 2019). There are different types of molecular interactions, and this paper mainly focuses on the ones that exist in bimolecular complexes, consisting of proteins, small molecules or RNA/DNAs. Specifically, to better capture their physical effects, we study molecular interactions via 3D geometry where atom coordinates are always provided.

Modeling molecular interaction relies heavily on how to represent molecules appropriately. In recent studies, Graph Neural Networks (GNNs) are applied for this purpose Gilmer et al. (2017); Jin et al. (2018). This is motivated by the fact that graphs naturally represent molecules, by considering atoms as nodes and inter-atom interactions or bonds as edges. When further encapsulating 3D atom coordinates, geometric graphs Gasteiger et al. (2020b); Schütt et al. (2017); Stärk et al. (2022) are used in place of conventional graph modeling that solely encodes topology. To process geometric graphs, equivariant GNNs, a new kind of GNNs that meet E(3) equivariance regarding translation, rotation and reflection are proposed, which exhibit promising performance in molecule interaction tasks Kong et al. (2022b); Luo et al. (2022); Townshend et al. (2020); Zhang et al. (2022).

Despite the encouraging progress, there still lacks a desirable and unified form of cross-domain molecular representation in molecular interaction. The molecules of different domains like small molecules, proteins, and RNA/DNAs are usually represented in different granularity, which consist of atoms, residues, and nucleobases, respectively. Existing approaches typically design domain-specific representations and model each of the interacting instances independently (Somnath et al., 2021; Wang et al., 2022), which are defective in learning the universal underlying interaction physics. Therefore, designing unified cross-domain molecular representation is demanded, which, however, is non-trivial. For one thing, directly applying unshared block-level graphs, whose nodes correspond to domain-specific building blocks, leads to limited transferability of the model from

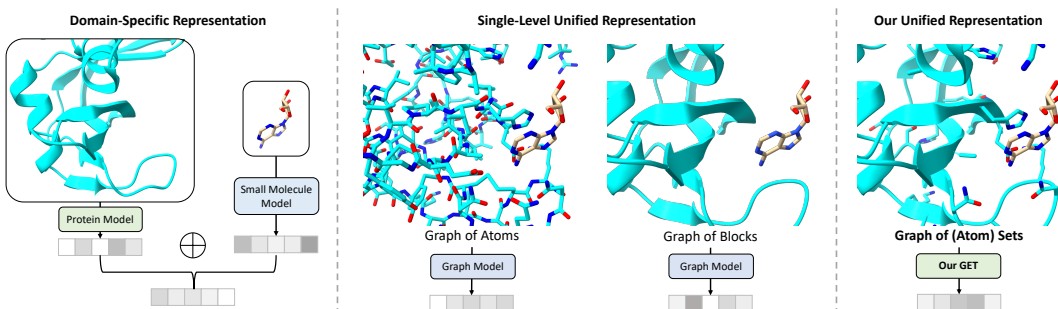

Figure 1: Domain-specific representations and unified representations in molecular interaction.

one domain to another. For another, decomposing all molecules into atom-level graphs discards the block specificity (*e.g.* which residue each atom belongs to) and overlook valuable heuristics for representation learning. It is still an open problem in designing a universal representation and a generalist model thereon to capture both the block-level specificity and atom-level shareability.

In this paper, we tackle this problem by modeling a complex involved in molecular interaction as *a geometric graph of sets*. This representation follows a bilevel design: in the top level, a complex is represented as a geometric graph of blocks; in the bottom level, each block contains a set of atomic instances. It is nontrivial to process such bilevel geometric graphs, as the model should handle blocks of variable sizes and ensure certain specific geometries. To this end, we propose *Generalist Equivariant Transformer (GET)*, which consists of the three modules: bilevel attention module, feed-forward module and layer normalization module. To be specific, the bilevel attention module updates the information of each atom by adopting both sparse block-level and dense atom-level attentions. The feed-forward module is to inject the intra-block geometry to each atom, and the layer normalization module is proposed to stabilize and accelerate the training. All the modules are E(3)-equivariant regarding the 3D coordinates, permutation-invariant regarding all atoms within each block, and work regardless of the varying block size. We compare our method with other representation approaches in Figure 1.

Notably, our formulation of graph of sets is relevant to conventional pooling-based hierarchical models based on graph of graphs (Jin et al., 2022). Nevertheless, these hierarchical architecture are usually inefficient and will blot out the fine-grained information after certain pooling-based aggregation, while our GET is able to retain both the atom-level and block-level information.

We conduct experiments on various molecular interactions between proteins, small molecules and RNA/DNAs. The results exhibit the superiority of our GET on the proposed unified representation over traditional methods including domain-specific independent models, single-level unified representations and hierarchical models. More excitingly, we identify strong potential of GET in capturing and transferring universal knowledge across different domains, and enabling zero-shot performance on RNA/DNA-ligand binding affinity prediction.

## 2 RELATED WORK

**Molecular Interaction and Representation** Various types of molecules across different domains (Du et al., 2016; Elfiky, 2020; Jones & Thornton, 1996) can form interactions, the strength of which are usually measured by the energy gap between the unbound and bound states of the molecules (*i.e.* affinity) (Gilson et al., 1997). We primarily investigate interactions between two proteins (Jones & Thornton, 1996) and between a protein and a small molecule (Du et al., 2016), both of which are widely explored in the machine learning community (Kong et al., 2022a; Luo et al., 2023; 2022; Somnath et al., 2021; Stärk et al., 2022; Wang et al., 2022). Furthermore, we embark on a pioneering effort to involve RNA/DNAs, which is a challenging endeavor due to the limited availability of such data. Small molecules are usually represented by graphs where nodes are atoms (Atz et al., 2021; Hoogeboom et al., 2022; Xu et al., 2022; Zaidi et al., 2022), but there are also explorations on subgraph-level decomposition of molecules by mining motifs (Geng et al., 2023; Jin et al., 2018; Kong et al., 2022b). Proteins are built upon residues, which are predefined sets of atoms (Richardson, 1981), and thus have mainly two categories of representations according to the granularity of graph nodes: atom-level and residue-level. Atom-level representations, as

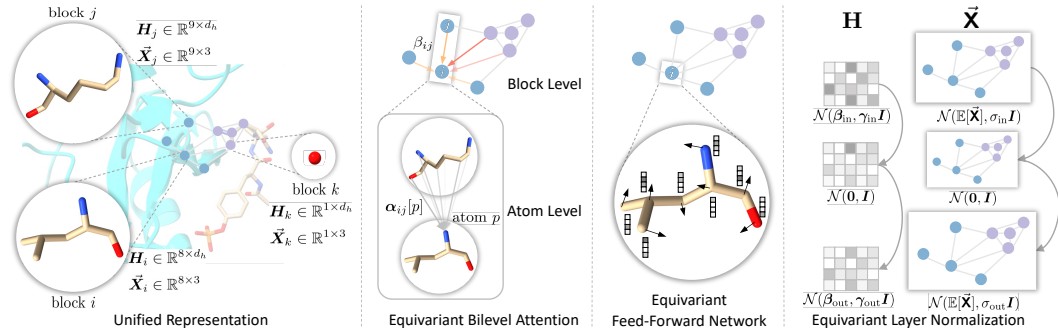

Figure 2: Overview of the unified representation and the equivariant modules in our Generalist Equivariant Transformer (GET). From left to right: The unified representation treats molecules as geometric graphs of sets according to predefined building blocks; The bilevel attention module captures both sparse block-level and dense atom-level interactions via an equivariant attention mechanism; The feed-forward network injects the block-level information into the intra-block atoms; The layer normalization transforms the input distribution with trainable scales and offsets.

the name suggests, decompose proteins into single atoms (Townshend et al., 2020) and discard the hierarchy of proteins. Residue-level representations either exert pooling on the atoms (Jin et al., 2022), or directly use residue-specific features (Anand & Achim, 2022; Shi et al., 2022; Somnath et al., 2021; Wang et al., 2022) which is limited to proteins. Similarly, RNA/DNAs also have atom-level and nucleobase-level representations (Avsec et al., 2021; Watson & Crick, 1953). Despite the differences in building blocks, the basic units (*i.e.* atoms) are shared across different molecular domains, and so do the fundamental interaction physics. Therefore, it is valuable to construct a unified representation for different molecular domains, which is explored in this paper.

**Equivariant Network** Equivariant networks integrate the symmetry of 3D world, namely E(3)-equivariance, into the models, and thus are widely used in geometric learning (Han et al., 2022). A line of methods rely on preprocessing 3D coordinates into invariant features, *e.g.* pairwise distances (Schütt et al., 2017; Choukroun & Wolf, 2021), angles (Gasteiger et al., 2020b;a; 2021; Liu et al., 2021b), to achieve invariant outputs. More recent works also keep track of equivariant features to achieve stronger expressivity (Joshi et al., 2023), either via scalarization (Schütt et al., 2021; Thölke & De Fabritiis, 2022; Du et al., 2023) or irreducible representations (Thomas et al., 2018; Batzner et al., 2022; Liao & Smidt, 2022; Batatia et al., 2022; Musaelian et al., 2023). Our work is inspired by multi-channel equivariant graph neural networks (Huang et al., 2022; Kong et al., 2022b) which assign each node with a coordinate matrix. However, they require a fixed number of channels (*i.e.* constant number of rows in the coordinate matrix) and lack invariance w.r.t to the permutations of the coordinates, which limits their application here as each building block is an unordered set of atoms with variable size. Moreover, the node features are still limited to single vector form (Thölke & De Fabritiis, 2022; Liao & Smidt, 2022), which is unable to accommodate all-atom representations in single blocks. In contrast, our proposed model is designed to handle geometric graphs of sets where each node contains an unordered set of 3D instances with a different size, which fits perfectly with the concept of building blocks in molecules.

## 3 METHOD

We start by illustrating the proposed unified representation for molecules in § 3.1. Then we introduce GET in § 3.2. Each layer of GET consists of the three types of E(3)-equivariant modules: a bilevel attention module, a feed-forward module, and a layer normalization after each previous module. The overall concepts are depicted in Figure 2 and the detailed scheme is presented in Appendix B.

### 3.1 UNIFIED REPRESENTATION: GEOMETRIC GRAPHS OF SETS

Graphs come as a central tool for molecular representations, and different kind of graphs is applied in different case. For instance, small molecules can be represented as single-level graphs, where each node is an atom, while proteins (RNA/DNAs) correspond to two-level graphs, where each

node is a residue (nucleobase) that consists of a variable number of atoms. To better characterize the interaction between different molecules, below we propose a unified molecular representation.

Given a complex consisting of a set of atoms $\mathbb{A}$, we first identify a set of blocks (*i.e.* subsets) from $\mathbb{A}$ according to some predefined notions (*e.g.* residues for proteins). Then the complex is abstracted as a geometric graph of sets $\mathcal{G} = (\mathcal{V}, \mathcal{E})$, where $\mathcal{V} = \{(\boldsymbol{H}_i, \vec{\boldsymbol{X}}_i) | 1 \leq i \leq B\}$ includes all $B$ blocks and $\mathcal{E} = \{(i, j, \boldsymbol{e}_{ij}) | 1 \leq i, j \leq B\}$ includes all edges between blocks[1], where $\boldsymbol{e}_{ij} \in \mathbb{R}^{d_e}$ distinguishes the type of the edge as intra-molecular or inter-molecular connection. In each block composed of $n_i$ atoms, $\boldsymbol{H}_i \in \mathbb{R}^{n_i \times d_h}$ denotes a set of atom feature vectors and $\vec{\boldsymbol{X}}_i \in \mathbb{R}^{n_i \times 3}$ denotes a set of 3D atom coordinates. To be specific, the $p$-th row of $\boldsymbol{H}_i$, which is the feature vector of atom $p$, sums up the trainable embeddings of atom types $\boldsymbol{a}_i[p]$, block types $b_i$, and atom position codes $\boldsymbol{p}_i[p]$ (see Appendix A), namely, $\boldsymbol{H}_i[p] = \text{Embed}(\boldsymbol{a}_i[p]) + \text{Embed}(b_i) + \text{Embed}(\boldsymbol{p}_i[p]), 1 \leq p \leq n_i$. To reduce the computational complexity, we construct $\mathcal{E}$ via k-nearest neighbors ($k = 9$) according to the block distance which is defined as the minimum distance between inter-block atom pairs:

$$d(i, j) = \min\{\|\vec{\boldsymbol{X}}_i[p] - \vec{\boldsymbol{X}}_j[q]\|_2 \mid 1 \leq p \leq n_i, 1 \leq q \leq n_j\}. \tag{1}$$

Overall, the block-level geometry derived from atomic interactions defines the connectivity of the graph, while the atom-level instances compose the unordered matrix-form node features with variable sizes. As we will observe in the next section, the above bilevel design allows our model to capture sparse interactions for the top level and dense interactions for the bottom level, achieving a desirable integration of different granularities. We will also demonstrate in the experiments that the representation can be easily extended to arbitrary block definitions (*e.g.* subgraph-level decomposition of small molecules (Kong et al., 2022b; Geng et al., 2023)).

**Connection to Single-Level Representations** If we restrict the blocks to one-atom subsets only, then we obtain the **atom-level** representation where each node is one atom, and correspondingly both $\boldsymbol{H}_i$ and $\vec{\boldsymbol{X}}_i$ are downgraded to row vectors as $n_i \equiv 1$. If we retain the building blocks but replace $\boldsymbol{H}_i$ and $\vec{\boldsymbol{X}}_i$ with their centroids, then we obtain the **block-level** representation where the atoms in the same block are pooled into one single instance. Both single-level representations assign a vector and a 3D coordinate to each node, hence can be fed into most structural learning models (Gasteiger et al., 2020b; Satorras et al., 2021; Schütt et al., 2017; Thölke & De Fabritiis, 2022). On the contrary, the proposed bilevel representation requires the capability of processing E(3)-equivariant feature matrices ($\boldsymbol{H}_i$ and $\vec{\boldsymbol{X}}_i$) with a variable number of rows, which cannot be directly processed by existing models. Additionally, within each block, the rows of $\boldsymbol{H}_i$ and $\vec{\boldsymbol{X}}_i$ are indeed elements in a set and their update should be unaffected by the row order. Luckily, the above challenges are well handled by our Generalist Equivariant Transformer proposed in the next section.

**Comparison with Hierarchical Representations** Previous studies (Jin et al., 2022) model proteins in a hierarchical manner, where the atom-level features within each residue are first pooled as the residue-level features that will be processed via the message passing over the graph of residues. In contrast to these hierarchical methods, our bilevel representation retains the information of both the atom-level and residue-level features simultaneously for attention-based message passing.

### 3.2 GENERALIST EQUIVARIANT TRANSFORMER

Upon the unified representation, we propose GET to model the structure of the input complex. As mentioned above, one beneficial property of GET is that it can tackle blocks of variable size. Besides, GET is sophisticatedly designed to ensure E(3)-equivariance and intra-block permutation invariance to handle the symmetry. Specifically, each layer of GET first exploits an equivariant bilevel attention module to capture both sparse interactions in block level and dense interactions in atom level. Then an equivariant feed-forward module updates each atom with the geometry of its block. Finally, a novel equivariant layer normalization is implemented on both the hidden states and coordinates. We present a detailed scheme of GET in Appendix B for better understanding.

**Equivariant Bilevel Attention Module** Given two blocks $i$ and $j$ of $n_i$ and $n_j$ atoms, respectively, we first obtain the query, the key, and the value matrices as follows:

$$\boldsymbol{Q}_i = \boldsymbol{H}_i \boldsymbol{W}_Q, \qquad \boldsymbol{K}_j = \boldsymbol{H}_j \boldsymbol{W}_K, \qquad \boldsymbol{V}_j = \boldsymbol{H}_j \boldsymbol{W}_V, \tag{2}$$

---

[1]We have added self-loops to reflect self-interactions between the atoms in each block.

where $\boldsymbol{W}_Q, \boldsymbol{W}_K, \boldsymbol{W}_V \in \mathbb{R}^{d_h \times d_r}$ are trainable parameters. We denote $\vec{\boldsymbol{X}}_{ij} \in \mathbb{R}^{n_i \times n_j \times 3}$ and $\boldsymbol{D}_{ij} \in \mathbb{R}^{n_i \times n_j}$ as the relative coordinates and distances between any atom pair in block $i$ and $j$, namely, $\vec{\boldsymbol{X}}_{ij}[p, q] = \vec{\boldsymbol{X}}_i[p] - \vec{\boldsymbol{X}}_j[q]$, $\boldsymbol{D}_{ij}[p, q] = \|\vec{\boldsymbol{X}}_{ij}[p, q]\|_2$.

The **atom-level cross attention values** from $j$ to $i$ are calculated by:

$$\boldsymbol{R}_{ij}[p, q] = \phi_A(\boldsymbol{Q}_i[p], \boldsymbol{K}_j[q], \mathrm{RBF}(\boldsymbol{D}_{ij}[p, q]), \boldsymbol{e}_{ij}), \tag{3}$$

$$\boldsymbol{\alpha}_{ij} = \mathrm{Softmax}(\boldsymbol{R}_{ij}\boldsymbol{W}_A). \tag{4}$$

Here, $\boldsymbol{e}_{ij}$ is the optional edge feature to distinguish between intra-molecule edges and inter-molecule edges; $\phi_A$ is a 2-layer Multi-Layer Perceptron (MLP) with SiLU (Hendrycks & Gimpel, 2016) activation; RBF (Gasteiger et al., 2020b) embeds the distance with radial basis functions (definition in Appendix B); $\boldsymbol{R}_{ij} \in \mathbb{R}^{n_i \times n_j \times d_r}$ represents the relations between each atom pair in block $i$ and $j$, which are later mapped to scalars by $\boldsymbol{W}_A \in \mathbb{R}^{d_r \times 1}$ to obtain the atom-level cross attentions $\boldsymbol{\alpha}_{ij} \in \mathbb{R}^{n_i \times n_j}$ between the two blocks through Softmax alone the columns of $\boldsymbol{R}_{ij}\boldsymbol{W}_A \in \mathbb{R}^{n_i \times n_j}$.

The **block-level cross attention value** from $j$ to $i$ is given by:

$$\boldsymbol{r}_{ij} = \frac{1}{n_i n_j} \sum_{p=1}^{n_i} \sum_{q=1}^{n_j} \boldsymbol{R}_{ij}[p, q], \tag{5}$$

$$\beta_{ij} = \frac{\exp(\boldsymbol{r}_{ij}\boldsymbol{W}_B)}{\sum_{j \in \mathcal{N}(i)} \exp(\boldsymbol{r}_{ij}\boldsymbol{W}_B)}, \tag{6}$$

where $\boldsymbol{W}_B \in \mathbb{R}^{d_r \times 1}$, and $\mathcal{N}(i)$ denotes the neighborhood blocks of $i$. Basically, $\boldsymbol{r}_{ij} \in \mathbb{R}^{d_r}$ represents the global relation between $i$ and $j$ after aggregating all values in $\boldsymbol{R}_{ij}$, which is then mapped to a scalar to obtain the block-level cross attentions $\beta_{ij}$ through Softmax in the neighborhood of $i$.

With the atom-level and the block-level attentions at hand, we are ready to update both the hidden states and coordinates for each atom $p$ in block $i$:

$$\boldsymbol{m}_{ij,p} = \boldsymbol{\alpha}_{ij}[p] \cdot \phi_v(\boldsymbol{V}_j \,\|\, \mathrm{RBF}(\boldsymbol{D}_{ij}[p])) \tag{7}$$

$$\vec{\boldsymbol{m}}_{ij,p} = \boldsymbol{\alpha}_{ij}[p] \cdot (\vec{\boldsymbol{X}}_{ij}[p] \odot \sigma_v(\boldsymbol{V}_j \,\|\, \mathrm{RBF}(\boldsymbol{D}_{ij}[p]))) \tag{8}$$

$$\boldsymbol{H}_i'[p] = \boldsymbol{H}_i[p] + \sum_{j \in \mathcal{N}(i)} \beta_{ij} \phi_m(\boldsymbol{m}_{ij,p}), \tag{9}$$

$$\vec{\boldsymbol{X}}_i'[p] = \vec{\boldsymbol{X}}_i[p] + \sum_{j \in \mathcal{N}(i)} \beta_{ij} (\sigma_m(\boldsymbol{m}_{ij,p}) \cdot \vec{\boldsymbol{m}}_{ij,p}), \tag{10}$$

where, $\|$ specifies the concatenation along the second dimension; $\phi_v, \phi_m, \sigma_v$, and $\sigma_m$ are all MLPs, $\phi_v$ and $\sigma_v$ are applied for each row of the input matrix independently; $\odot$ computes the element-wise multiplication. It is verified that the shape of the updated variables $\boldsymbol{H}_i'$ and $\vec{\boldsymbol{X}}_i'$ keeps the same irregardless of the value of the block size $n_j$. In addition, since the attentions $\boldsymbol{\alpha}_{ij}$ and $\beta_{ij}$ are E(3)-invariant, the update of $\vec{\boldsymbol{X}}_i'$ is E(3)-equivariant. It can also be observed that the update is independent to the atom permutation of each block. We provide detailed proofs in Appendix C.

**Equivariant Feed-Forward Network**  This module updates $\boldsymbol{H}_i$ and $\vec{\boldsymbol{X}}_i$ for each atom individually. We denote each row of $\boldsymbol{H}_i$ as $\boldsymbol{h}$, and $\vec{\boldsymbol{X}}_i$ as $\vec{\boldsymbol{x}}$. We first calculate the centroids of the block:

$$\boldsymbol{h}_c = \mathrm{centroid}(\boldsymbol{H}_i), \qquad \vec{\boldsymbol{x}}_c = \mathrm{centroid}(\vec{\boldsymbol{X}}_i). \tag{11}$$

Then we obtain the relative coordinate $\Delta\vec{\boldsymbol{x}}$ as well as the distance representation $\boldsymbol{r}$ between each atom and the centroid:

$$\Delta\vec{\boldsymbol{x}} = \vec{\boldsymbol{x}} - \vec{\boldsymbol{x}}_c, \qquad \boldsymbol{r} = \mathrm{RBF}(\|\Delta\vec{\boldsymbol{x}}\|_2), \tag{12}$$

The centroids and the distance representation are then integrated into the updating process of $\boldsymbol{h}$ and $\vec{\boldsymbol{x}}$ to let each atom be aware of the geometric context of its block, where $\phi_h, \sigma_x$ are MLPs:

$$\boldsymbol{h}' = \boldsymbol{h} + \phi_h(\boldsymbol{h}, \boldsymbol{h}_c, \boldsymbol{r}), \tag{13}$$

$$\vec{\boldsymbol{x}}' = \vec{\boldsymbol{x}} + \Delta\vec{\boldsymbol{x}}\sigma_x(\boldsymbol{h}, \boldsymbol{h}_c, \boldsymbol{r}), \tag{14}$$

**Equivariant Layer Normalization**   Layer normalization is known to stabilize and accelerate the training of deep neural networks (Ba et al., 2016; Vaswani et al., 2017). The challenge here is that we need to additionally consider E(3)-equivariance when normalizing the coordinates. To this end, we first extract the centroid of the entire graph as $\mathbb{E}[\vec{\mathbf{X}}]$, where $\vec{\mathbf{X}}$ collects the coordinates of all atoms in all blocks. Then we exert layer normalization on the hidden vectors and coordinates of individual atoms as follows:

$$\boldsymbol{h}' = \frac{\boldsymbol{h} - \mathbb{E}[\boldsymbol{h}]}{\sqrt{\mathrm{Var}[\boldsymbol{h}]}} \cdot \boldsymbol{\gamma} + \boldsymbol{\beta}, \tag{15}$$

$$\vec{\boldsymbol{x}}' = \frac{\vec{\boldsymbol{x}} - \mathbb{E}[\vec{\mathbf{X}}]}{\sqrt{\mathrm{Var}[\vec{\mathbf{X}} - \mathbb{E}[\vec{\mathbf{X}}]]}} \cdot \sigma + \mathbb{E}[\vec{\mathbf{X}}], \tag{16}$$

where $\boldsymbol{\gamma}, \boldsymbol{\beta}$, and $\sigma$ are learnable parameters, and $\mathrm{Var}[\vec{\mathbf{X}}]$ calculates the variation of all atom coordinates with respect to the centroid. Therefore, the coordinates, after subtracting the centroid of all atoms, are first normalized to standard Gaussian distribution and then scaled with $\sigma$ before recovering the centroid. In addition, to further reflect the rescaling of the coordinates into hidden features, we inject the following update before applying the above layer normalization:

$$\boldsymbol{h} = \boldsymbol{h} + \phi_{\mathrm{LN}}(\mathrm{RBF}(\sigma/\sqrt{\mathrm{Var}[\vec{\mathbf{X}}]})), \tag{17}$$

where $\phi_{\mathrm{LN}}$ is an MLP. In contrast to existing literature which only implements layer normalization on E(3)-invariant features (Thölke & De Fabritiis, 2022; Liao & Smidt, 2022) or node-wise velocities (Zaidi et al., 2022), ours works on both E(3)-invariant features and E(3)-equivariant coordinates.

Thanks to the E(3)-equivariance of each module, GET, which is the cascading of these modules in each layer, also conforms to the symmetry of the 3D world. We provide the proof in Appendix C and complexity analysis in Appendix D.

## 4   EXPERIMENTS

In this section, we aim to answer the following three questions via empirical experiments: (1) Does modeling complexes with unified representation better captures the geometric interactions than treating each interacting entity independently with domain-specific representations (§ 4.1)? (2) Is the proposed unified representation more expressive than vanilla single-level representations or pooling-based hierarchical methods (§ 4.2)? (3) Can the proposed method generalize to different domains by learning the universal underlying interaction physics (§ 4.3)?

We conduct experiments on prediction of binding between proteins, small molecules and nucleic acids. Thus, we adopt three widely used metrics for quantitive evaluation (Townshend et al., 2020; Liu et al., 2021a; Luo et al., 2023; Notin et al., 2022): **RMSE** is the Root Mean Square Error of the predicted value; **Pearson Correlation** (Cohen et al., 2009) measures the linear correlation between the predicted values and the target values; **Spearman Correlation** (Hauke & Kossowski, 2011) measures the correlation between the rankings given by the predicted and the target values

### 4.1   COMPARISON TO DOMAIN-SPECIFIC REPRESENTATIONS

We evaluate our method on the prediction of binding affinity between proteins and small molecules against state-of-the-art two-branch models with domain-specific representations from existing literature (Somnath et al., 2021; Wang et al., 2022). We follow Somnath et al. (2021); Wang et al. (2022) to conduct experiments on the well-established PDBbind (Wang et al., 2004; Liu et al., 2015) and split the dataset (4,709 biomolecular complexes) according to sequence identity of the protein with 30% as the threshold. Details of the experiments are provided in Appendix F.

**Results**   Table 1 shows that our GET surpasses the baselines by a large margin. Compared to the baselines which encode proteins and small molecules independently with delicately designed domain-specific models, our unified representation enables unified geometric learning with only one model, which better captures the interactive geometric information between the protein and the small molecule. Notably, among models with one encoder (Jing et al., 2021; Townshend et al.,

Table 1: The mean and the standard deviation of three runs on the PDBbind benchmark. The best results are marked in bold and the second best are underlined. The results of baselines are borrowed from Wang et al. (2022). Baselines encoding the complexes with one model are marked with $*$.

| Model | RMSE↓ | Pearson↑ | Spearman↑ |
|---|---|---|---|
| DeepDTA (Öztürk et al., 2018) | $1.866 \pm 0.080$ | $0.472 \pm 0.022$ | $0.471 \pm 0.024$ |
| Bepler and Berger (Bepler & Berger, 2019) | $1.985 \pm 0.006$ | $0.165 \pm 0.006$ | $0.152 \pm 0.024$ |
| TAPE (Rao et al., 2019) | $1.890 \pm 0.035$ | $0.338 \pm 0.044$ | $0.286 \pm 0.124$ |
| ProtTrans (Elnaggar et al., 2022) | $1.544 \pm 0.015$ | $0.438 \pm 0.053$ | $0.434 \pm 0.058$ |
| MaSIF (Gainza et al., 2020) | $1.484 \pm 0.018$ | $0.467 \pm 0.020$ | $0.455 \pm 0.014$ |
| IEConv (Hermosilla et al., 2020) | $1.554 \pm 0.016$ | $0.414 \pm 0.053$ | $0.428 \pm 0.032$ |
| Holoprot-Full Surface (Somnath et al., 2021) | $1.464 \pm 0.006$ | $0.509 \pm 0.002$ | $0.500 \pm 0.005$ |
| Holoprot-Superpixel (Somnath et al., 2021) | $1.491 \pm 0.004$ | $0.491 \pm 0.014$ | $0.482 \pm 0.032$ |
| ProNet-Amino Acid (Wang et al., 2022) | $1.455 \pm 0.009$ | $0.536 \pm 0.012$ | $0.526 \pm 0.012$ |
| ProtNet-Backbone (Wang et al., 2022) | $1.458 \pm 0.003$ | $0.546 \pm 0.007$ | $0.550 \pm 0.008$ |
| ProtNet-All-Atom (Wang et al., 2022) | $1.463 \pm 0.001$ | $\underline{0.551 \pm 0.005}$ | $0.551 \pm 0.008$ |
| GVP$^*$ (Jing et al., 2021) | $1.594 \pm 0.073$ | - | - |
| Atom3D-3DCNN$^*$ (Townshend et al., 2020) | $\underline{1.416 \pm 0.021}$ | $0.550 \pm 0.021$ | $\underline{0.553 \pm 0.009}$ |
| Atom3D-ENN$^*$ (Townshend et al., 2020) | $1.568 \pm 0.012$ | $0.389 \pm 0.024$ | $0.408 \pm 0.021$ |
| Atom3D-GNN$^*$ (Townshend et al., 2020) | $1.601 \pm 0.048$ | $0.545 \pm 0.027$ | $0.533 \pm 0.033$ |
| GET$^*$ (ours) | $\mathbf{1.364 \pm 0.009}$ | $\mathbf{0.596 \pm 0.006}$ | $\mathbf{0.573 \pm 0.007}$ |

2020), our method also achieves significant improvement since our unified representation retains domain-specific hierarchies instead of decomposing all types of molecules into atomic graphs.

## 4.2 COMPARISON TO VANILLA UNIFIED REPRESENTATIONS

Next, we compare the proposed unified representation with three vanilla unified representations: (1) **Block**-level methods assign each building block to one node where the definition of building block is domain-specific (*e.g.* each residue in the proteins is one node); (2) **Atom**-level methods treats all kinds of molecules as graphs of atoms; (3) **Hierarchical** methods first implement message passing on atom-level graphs, then obtain the block-level representations by pooling for further message passing on the block-level graphs (Jin et al., 2022).

**Baselines** These vanilla unified representations are compatible with most geometric graph models in existing literature, thus we adopt the following representative models as backbone for comparison. **SchNet** (Schütt et al., 2017), **DimeNet++** (Gasteiger et al., 2020b;a), and **GemNet** (Gasteiger et al., 2021) build invariant models based on invariant geometric features (*i.e.* distances and angles). **EGNN** (Satorras et al., 2021), TorchMD-Net (**ET**) (Thölke & De Fabritiis, 2022), and **LEFT-Net** (Du et al., 2023) keep track of equivariant features and are implemented directly on 3D coordinates via scalarization (Han et al., 2022). **MACE** (Batatia et al., 2022) and **Equiformer** (Liao & Smidt, 2022) leverage spherical harmonics and irreducible representations (Thomas et al., 2018) to compose equivariant models.

**Dataset** To this end, we evaluate the models on prediction of protein-protein affinity and ligand-binding affinity. For **Protein-Protein Affinity** (PPA), we adopt the Protein-Protein Affinity Benchmark Version 2 (Kastritis et al., 2011; Vreven et al., 2015) as the test set, which categorizes 176 diversified protein-protein complexes into three difficulty levels (*i.e.* Rigid, Medium, Flexible) according to the conformation change of the proteins from the unbound to the bound state (Kastritis et al., 2011). The Flexible split is the most challenging as the proteins undergo large conformation change upon binding. As for training, we obtain 2,500 complexes with annotated binding affinity ($K_i$ or $K_d$) from PDBbind (Wang et al., 2004) and split the dataset according to sequence identity on a threshold of 30%. For **Ligand-Binding Affinity** (LBA), we use the LBA dataset and its splits in Atom3D benchmark (Townshend et al., 2020), where there are 3507, 466, and 490 complexes in the training, the validation, and the test sets. Details are provided in Appendix F.

**Results** We report the mean and the standard deviation of the metrics across three runs for PPA and LBA in Table 2. Details on different difficulty levels for PPA are included in Appendix I due to the space limit. Inspiringly, it reads that our GET with the proposed unified representation achieves significantly better performance compared with the baselines with either single-level representations or hierarchical pooling, no matter the interacting partners are macro molecules (*i.e.* proteins) or small molecules. This confirms the superiority of our method, which comes from a desirable integration

Table 2: The mean and the standard deviation of three runs on PPA and LBA prediction. The best results are marked in bold and the second best are underlined. Baselines that fail to process atomic graphs due to high complexity are marked with OOM (out of memory).

| Repr. | Model | PPA | | LBA | | |
|---|---|---|---|---|---|---|
| | | Pearson↑ | Spearman↑ | RMSE↓ | Pearson↑ | Spearman↑ |
| Block | SchNet | $0.439 \pm 0.016$ | $0.427 \pm 0.012$ | $1.406 \pm 0.020$ | $0.565 \pm 0.006$ | $0.549 \pm 0.007$ |
| | DimeNet++ | $0.323 \pm 0.025$ | $0.317 \pm 0.031$ | $1.391 \pm 0.020$ | $0.576 \pm 0.016$ | $0.569 \pm 0.016$ |
| | EGNN | $0.381 \pm 0.021$ | $0.382 \pm 0.022$ | $1.409 \pm 0.015$ | $0.566 \pm 0.010$ | $0.548 \pm 0.012$ |
| | ET | $0.424 \pm 0.021$ | $0.415 \pm 0.027$ | $1.367 \pm 0.037$ | $0.599 \pm 0.017$ | $0.584 \pm 0.025$ |
| | GemNet | $0.387 \pm 0.023$ | $0.393 \pm 0.027$ | $1.393 \pm 0.036$ | $0.569 \pm 0.027$ | $0.553 \pm 0.026$ |
| | MACE | $0.470 \pm 0.015$ | $0.466 \pm 0.011$ | $1.385 \pm 0.006$ | $0.599 \pm 0.010$ | $0.580 \pm 0.014$ |
| | Equiformer | $\underline{0.484 \pm 0.007}$ | $\underline{0.496 \pm 0.007}$ | $\underline{1.350 \pm 0.019}$ | $\underline{0.604 \pm 0.013}$ | $\underline{0.591 \pm 0.012}$ |
| | LEFTNet | $0.452 \pm 0.013$ | $0.452 \pm 0.013$ | $1.377 \pm 0.013$ | $0.588 \pm 0.011$ | $0.576 \pm 0.010$ |
| Atom | SchNet | $0.369 \pm 0.007$ | $0.404 \pm 0.016$ | $1.357 \pm 0.017$ | $0.598 \pm 0.011$ | $0.592 \pm 0.015$ |
| | DimeNet++ | OOM | OOM | $1.439 \pm 0.036$ | $0.547 \pm 0.015$ | $0.536 \pm 0.016$ |
| | EGNN | $0.302 \pm 0.010$ | $0.349 \pm 0.009$ | $1.358 \pm 0.000$ | $0.599 \pm 0.002$ | $0.587 \pm 0.004$ |
| | ET | $0.401 \pm 0.005$ | $0.436 \pm 0.004$ | $1.381 \pm 0.013$ | $0.591 \pm 0.007$ | $0.583 \pm 0.009$ |
| | GemNet | OOM | OOM | OOM | OOM | OOM |
| | MACE | $0.463 \pm 0.052$ | $0.449 \pm 0.052$ | $1.411 \pm 0.029$ | $0.579 \pm 0.009$ | $0.563 \pm 0.012$ |
| | Equiformer | OOM | OOM | OOM | OOM | OOM |
| | LEFTNet | $0.448 \pm 0.046$ | $0.431 \pm 0.046$ | $1.343 \pm 0.004$ | $0.610 \pm 0.004$ | $0.598 \pm 0.003$ |
| Hierarchical | SchNet | $0.438 \pm 0.017$ | $0.424 \pm 0.016$ | $1.370 \pm 0.028$ | $0.590 \pm 0.017$ | $0.571 \pm 0.028$ |
| | DimeNet++ | OOM | OOM | $1.388 \pm 0.010$ | $0.582 \pm 0.009$ | $0.574 \pm 0.007$ |
| | EGNN | $0.386 \pm 0.021$ | $0.390 \pm 0.016$ | $1.380 \pm 0.015$ | $0.586 \pm 0.004$ | $0.568 \pm 0.004$ |
| | ET | $0.401 \pm 0.005$ | $0.438 \pm 0.029$ | $1.383 \pm 0.009$ | $0.580 \pm 0.008$ | $0.564 \pm 0.004$ |
| | GemNet | OOM | OOM | OOM | OOM | OOM |
| | MACE | $0.466 \pm 0.020$ | $0.470 \pm 0.016$ | $1.372 \pm 0.021$ | $0.612 \pm 0.010$ | $0.592 \pm 0.010$ |
| | Equiformer | OOM | OOM | OOM | OOM | OOM |
| | LEFTNet | $0.445 \pm 0.024$ | $0.446 \pm 0.029$ | $1.366 \pm 0.016$ | $0.592 \pm 0.014$ | $0.580 \pm 0.011$ |
| Unified | GET (ours) | $\mathbf{0.514 \pm 0.011}$ | $\mathbf{0.533 \pm 0.011}$ | $1.327 \pm 0.005$ | $0.620 \pm 0.004$ | $0.611 \pm 0.003$ |
| | GET-PS (ours) | - | - | $\mathbf{1.309 \pm 0.012}$ | $\mathbf{0.633 \pm 0.008}$ | $\mathbf{0.642 \pm 0.009}$ |

of different granularities. Further, to show the flexibility of the proposed unified representation, as mentioned in § 3.1, we add **GET-PS**, which defines the blocks in small molecules as principal subgraphs (Kong et al., 2022b) instead of atoms. GET-PS receives obvious gains over GET since fragments in small molecules usually contribute to interactions as a whole (Hajduk & Greer, 2007).

## 4.3 GENERALIZATION ACROSS DIFFERENT DOMAINS

Finally, we explore whether our model is able to find universal underlying physics that can generalize across different domains.

**Data Augmentation from Different Domains** We mix the dataset of protein-protein affinity and protein-ligand affinity for training, and evaluate the models on the test set of the two domains, respectively. We also benchmark ET, MACE, and LEFTNet, which exhibit competitive performance and efficiency in § 4.2, under the same setting for comparison. We present the results in Figure 3, and include detailed mean and standard deviation in Appendix J. The results demonstrate that our method obtains benefits from the mixed training set on both PPA and LBA, while the baselines receive negative impact in most cases. These phenomena well demonstrate the generalization ability of the proposed GET equipped with the unified representation.

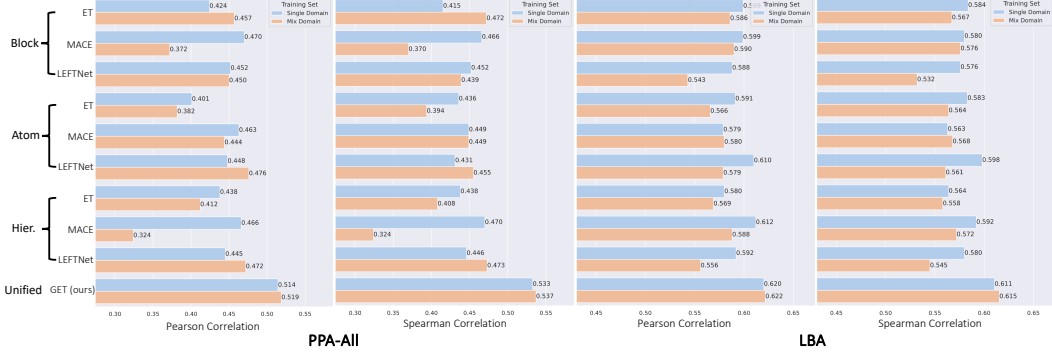

Figure 3: Comparison of different models on the universal learning of molecular interaction affinity.

**Zero-Shot Prediction of DNA/RNA-Ligand Affinity** A more practical and meaningful, yet also more challenging scenario is ligand (small molecule) binding on nucleic acids (RNA/DNA), the data of which are scarce and expensive to obtain. We use the 149 data points available in PDBbind (Wang

et al., 2004) as the zero-shot test set, and train a model on binding data from other domains in PDBbind (*i.e.* protein-protein, protein-ligand and RNA/DNA-protein). The results in Table 3 show that GET achieves amazing generalizability across different domains on molecular interaction.

Table 3: Zero-shot performance on DNA/RNA-ligand binding affinity prediction for three runs.

| Repr. | Model | Pearson↑ | Spearman↑ |
|---|---|---|---|
| Block | ET | $0.217 \pm 0.059$ | $0.185 \pm 0.051$ |
| | MACE | $0.004 \pm 0.045$ | $0.045 \pm 0.034$ |
| | LEFTNet | $0.279 \pm 0.127$ | $0.252 \pm 0.082$ |
| Atom | ET | $0.150 \pm 0.034$ | $0.198 \pm 0.043$ |
| | MACE | $-0.005 \pm 0.079$ | $0.027 \pm 0.083$ |
| | LEFTNet | $0.271 \pm 0.062$ | $0.279 \pm 0.062$ |
| Hierarchical | ET | $0.348 \pm 0.047$ | $0.302 \pm 0.028$ |
| | MACE | $0.002 \pm 0.055$ | $0.041 \pm 0.030$ |
| | LEFTNet | $0.279 \pm 0.122$ | $0.259 \pm 0.077$ |
| Unified | GET | $\mathbf{0.450 \pm 0.054}$ | $\mathbf{0.362 \pm 0.041}$ |

Both experiments confirm the potential of our model to discover universal underlying principles of molecular interactions capable of generalizing across diverse domains.

## 5 ANALYSIS

We conduct ablation study by removing the following modules: the layer normalization (w/o LN); the equivariant normalization on coordinates in the LN (w/o equivLN); the reflection of rescaling information in hidden features in Eq. 17 (w/o EmbedScale); the equivariant feed-forward network (w/o FFN); both LN and FFN (w/o LN & FFN). The results are presented in Table 4.

The ablations of the modules reveal following regularities: (1) Removing either the entire layer normalization or only the equivariant normalization on coordinates introduces instability in training, which not only leads to higher variance across different experiments, but also induces adverse impacts in some tasks like PPA; (2) Not reflecting the rescaling information in the hidden features has an adverse effect on the performance as the scale of the coordinates also carries essential information for learning the interaction physics; (3) The removal of the equivariant feed-forword module incurs detriment to the overall performance, indicating the necessity of the FFN to encourage intra-block geometrical communications between atoms. We include analysis on attention in Appendix K.

Table 4: Ablation study of each module in our proposed Generalist Equivariant Transformer (GET), where LN and FFN are abbreviations for LayerNorm and Feed-Forward Network, respectively. The best results are marked in bold and the second best are underlined.

| Repr. | Model | PPA-All | | LBA | |
|---|---|---|---|---|---|
| | | Pearson↑ | Spearman↑ | Pearson↑ | Spearman↑ |
| Unified | GET-mix | $\mathbf{0.519 \pm 0.004}$ | $\mathbf{0.537 \pm 0.003}$ | $\mathbf{0.622 \pm 0.006}$ | $\mathbf{0.615 \pm 0.008}$ |
| | GET | $\underline{0.514 \pm 0.011}$ | $\underline{0.533 \pm 0.011}$ | $\underline{0.620 \pm 0.004}$ | $\underline{0.611 \pm 0.003}$ |
| | w/o LN | $0.366 \pm 0.024$ | $0.426 \pm 0.032$ | $0.589 \pm 0.007$ | $0.593 \pm 0.008$ |
| | w/o equivLN | $0.368 \pm 0.025$ | $0.426 \pm 0.032$ | $0.591 \pm 0.008$ | $0.597 \pm 0.009$ |
| | w/o EmbedScale | $0.490 \pm 0.027$ | $0.507 \pm 0.030$ | $0.591 \pm 0.002$ | $0.586 \pm 0.003$ |
| | w/o FFN | $0.494 \pm 0.010$ | $0.510 \pm 0.010$ | $0.593 \pm 0.008$ | $0.601 \pm 0.012$ |
| | w/o LN & FFN | $0.360 \pm 0.018$ | $0.423 \pm 0.024$ | $0.589 \pm 0.009$ | $0.596 \pm 0.008$ |

## 6 CONCLUSION

In this paper, we explore the unified representation of molecules as geometric graphs of sets, which enables all-atom representations while preserving the heuristic building blocks of different molecules. To model the unified representaion, we propose a Generalist Equivariant Transformer (GET) to accommodate matrix-form node features and coordinates with E(3)-equivariance and permutation invariance. Each layer of GET consists of a bilevel attention module, a feed-forward module, and an equivariant layer normalization after each of the previous two modules. Experiments on molecular interactions demonstrate the superiority of learning unified representation with our GET compared to single-level representations and existing baselines. Further explorations on mixing molecular types reveal the ability of our method to learn generalizable molecular interaction mechanisms, which could inspire future research on universal representation learning of molecules.

## REPRODUCIBILITY

Our codes are available at `https://anonymous.4open.science/r/GET-anonymous/` with anonymized contents.

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

## A    ATOM POSITION CODE

Certain types of molecules have conventional position codes to distinguish different status of the atoms in the same block. For example, in the protein domain, where building blocks are residues, each atom in a residue is assigned a position code $(\alpha, \beta, \gamma, \delta, \varepsilon, \zeta, \eta, ...)$ according to the number of chemical bonds between it and the alpha carbon (i.e. $C_\alpha$). As these position codes provide meaningful heuristics of intra-block geometry, we also include them as a component of the embedding. For other types of molecules without such position codes (e.g. small molecules), we assign a [BLANK] type for positional embedding.

## B    SCHEME OF THE GENERALIST EQUIVARIANT TRANSFORMER

We depict the overall workflow and the details of the equivariant bilevel attention module in Figure 4.

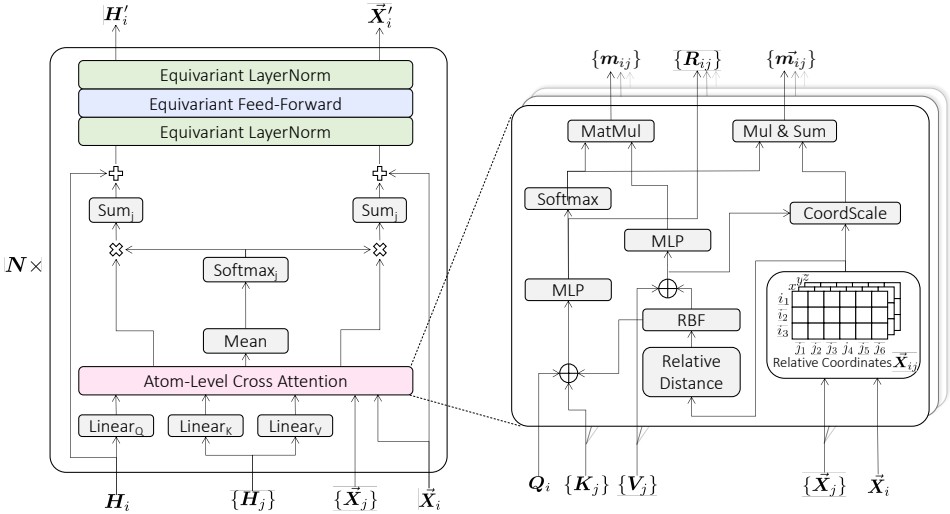

Figure 4: The scheme of a layer of Generalist Equivariant Transformer, where block $i$ $(\boldsymbol{H}_i, \vec{\boldsymbol{X}}_i)$ is updated by its neighbors $(\{\boldsymbol{H}_j\}, \{\vec{\boldsymbol{X}}_j\}, j \in \mathcal{N}_b(i))$. $\times$, $+$, and $\oplus$ denote multiplication, addition and concatenation, respectively. (Left) The overall workflow of a layer and details of the block-level attention. (Right) The details of the atom-level cross attention. GET is composed of $N$ such layers.

RBF embeds distances into $d_{\text{rbf}}$-dimensional vectors through radial basis functions:

$$\text{RBF}(d)_k = u(\frac{d}{c}) \cdot \exp(-\frac{|d - \mu_k|}{2\gamma}), 1 \le k \le d_{\text{rbf}} \tag{18}$$

where $\mu_k$ is uniformly distributed in $[0, c]$ ($c = 7.0$ in our paper), $\gamma = \frac{c}{d_{\text{rbf}}}$, and $u(d)$ is the polynomial envelope function for cutting off large distances (Gasteiger et al., 2020b) ($p = 5$ in our paper):

$$u(x) = 1 - \frac{(p+1)(p+2)}{2}x^p + p(p+2)x^{p+1} - \frac{p(p+1)}{2}x^{p+2} \tag{19}$$

## C    PROOF OF E(3)-EQUIVARIANCE AND INTRA-BLOCK PERMUTATION INVARIANCE

**Theorem C.1** (E(3)-Equivariance and Intra-Block Permutation Invariance). *Denote the proposed Equivariant Transformer as $\{\boldsymbol{H}_i', \vec{\boldsymbol{X}}_i'\} = \text{GET}(\{\boldsymbol{H}_i, \vec{\boldsymbol{X}}_i\})$, then it conforms to E(3)-Equivariance and Intra-Block Permutation Invariance. Namely, $\forall g \in E(3), \forall \{\pi_i \in S_{n_i} | 1 \le i \le B\}$, where $B$ is the number of blocks in the input and $S_{n_i}$ includes all permutations on $n_i$ elements, we have $\{\pi_i \cdot \boldsymbol{H}_i', \pi_i \cdot g \cdot \vec{\boldsymbol{X}}_i'\} = \text{GET}(\{\pi_i \cdot \boldsymbol{H}_i, \pi_i \cdot g \cdot \vec{\boldsymbol{X}}_i\})$.*

The Generalist Equivariant Transformer (GET) is the cascading of the three types of modules: bilevel attention, feed-forward network, and layer normalization. Further, the E(3)-equivariance

and the intra-block permutation invariance are disentangled. Therefore, the proof of its E(3)-equivariance and its invariance respect to the intra-block permutations can be decomposed into proof of these two properties on each module, which we present below.

## C.1 PROOF OF E(3)-EQUIVARIANCE

First we give the definition of E(3)-equivariance as follows:

**Definition C.2** (E(3)-equivariance). A function $\phi : \mathbb{X} \to \mathbb{Y}$ conforms E(3)-equivariance if $\forall g \in$ E(3), the equation $\rho_{\mathbb{Y}}(g)\boldsymbol{y} = \phi(\rho_{\mathbb{X}}(g)\boldsymbol{x})$ holds true, where $\rho_{\mathbb{X}}$ and $\rho_{\mathbb{Y}}$ instantiate $g$ in $\mathbb{X}$ and $\mathbb{Y}$, respectively. A special case is E(3)-invariance where $\rho_{\mathbb{Y}}$ constantly outputs identity transformation (i.e. $\rho_{\mathbb{Y}}(g) \equiv I$).

Given $g \in$ E(3) and $\vec{\boldsymbol{x}} \in \mathbb{R}^3$, we can instantiate $g$ as $g \cdot \vec{\boldsymbol{x}} := \boldsymbol{Q}\vec{\boldsymbol{x}} + \vec{\boldsymbol{t}}$, where $\boldsymbol{Q} \in \mathbb{R}^{3 \times 3}$ is an orthogonal matrix and $\vec{\boldsymbol{t}} \in \mathbb{R}^3$ is a translation vector. Implementing $g$ on a coordinate matrix $\vec{\boldsymbol{X}} \in \mathbb{R}^{n \times 3}$ means transforming each coordinate (i.e. each row) with $g$.

Then we prove the E(3)-equivariance of each module in GET as follows:

**Lemma C.3.** *Denote the bilevel attention module as $\{\boldsymbol{H}'_i, \vec{\boldsymbol{X}}'_i\} = \text{Att}(\{\boldsymbol{H}_i, \vec{\boldsymbol{X}}_i\})$, then it is E(3)-equivariant. Namely, $\forall g \in E(3)$, we have $\{\boldsymbol{H}'_i, g \cdot \vec{\boldsymbol{X}}'_i\} = \text{Att}(\{\boldsymbol{H}_i, g \cdot \vec{\boldsymbol{X}}_i\})$.*

*Proof.* The key to the proof of Lemma C.3 is to prove that the propagation in Eq. 2-10 is E(3)-invariant on $\boldsymbol{H}_i$ and E(3)-equivariant on $\vec{\boldsymbol{X}}_i$. Obviously, the correlation $\boldsymbol{R}_{ij}$ between block $i$ and block $j$ in Eq. 3 is E(3)-invariant because all the inputs, that is, the query, the key, and the distance matrices, are not affected by the geometric transformation $g$. Therefore, we can immediately derive that the atom-level cross attention $\boldsymbol{\alpha}_{ij}$ in Eq. 4 is E(3)-invariant. Similarly, the block-level attention $\beta_{ij}$ in Eq. 6 is E(3)-invariant because it only operates on $\boldsymbol{r}_{ij}$ in Eq. 5 which aggregates $\boldsymbol{\alpha}_{ij}$ and the edge feature. Finally, we can derive the E(3)-invariance on $\boldsymbol{H}$ and the E(3)-equivariance on $\vec{\boldsymbol{X}}$:

$$\boldsymbol{H}'_i[p] = \boldsymbol{H}_i[p] + \sum_{j \in \mathcal{N}(i)} \beta_{ij}\phi_m(\boldsymbol{m}_{ij,p}),$$

$$= \boldsymbol{H}_i[p] + \sum_{j \in \mathcal{N}(i)} \beta_{ij}\phi_m(\boldsymbol{\alpha}_{ij}[p] \cdot \phi_v(\boldsymbol{V}_j \parallel \text{RBF}(\boldsymbol{D}_{ij}[p]))),$$

$$g \cdot \vec{\boldsymbol{X}}'_i[p] = g \cdot \left( \vec{\boldsymbol{X}}_i[p] + \sum_{j \in \mathcal{N}(i)} \beta_{ij}(\vec{\boldsymbol{m}}_{ij,p} \odot \sigma_m(\boldsymbol{m}_{ij,p})) \right)$$

$$= g \cdot \left( \vec{\boldsymbol{X}}_i[p] + \sum_{j \in \mathcal{N}(i)} \beta_{ij}\left(\boldsymbol{\alpha}_{ij}[p] \cdot (\vec{\boldsymbol{X}}_{ij}[p] \odot \sigma_v(\boldsymbol{V}_j \parallel \text{RBF}(\boldsymbol{D}_{ij}[p]))) \odot \sigma_m(\boldsymbol{m}_{ij,p})\right) \right)$$

$$= \boldsymbol{Q}\left( \vec{\boldsymbol{X}}_i[p] + \sum_{j \in \mathcal{N}(i)} \beta_{ij}\left(\boldsymbol{\alpha}_{ij}[p] \cdot (\vec{\boldsymbol{X}}_{ij}[p] \odot \sigma_v(\boldsymbol{V}_j \parallel \text{RBF}(\boldsymbol{D}_{ij}[p]))) \odot \sigma_m(\boldsymbol{m}_{ij,p})\right) \right) + \vec{\boldsymbol{t}}$$

$$= (\boldsymbol{Q}\vec{\boldsymbol{X}}_i[p] + \vec{\boldsymbol{t}}) + \sum_{j \in \mathcal{N}(i)} \beta_{ij} \left( \boldsymbol{\alpha}_{ij}[p] \cdot \left( \begin{bmatrix} \boldsymbol{Q}(\vec{\boldsymbol{X}}_i[p] - \vec{\boldsymbol{X}}_j[1]) \\ \vdots \\ \boldsymbol{Q}(\vec{\boldsymbol{X}}_i[p] - \vec{\boldsymbol{X}}_j[n_j]) \end{bmatrix} \odot \sigma_v(\boldsymbol{V}_j \parallel \text{RBF}(\boldsymbol{D}_{ij}[p])) \right) \odot \sigma_m(\boldsymbol{m}_{ij,p}) \right)$$

$$= (\boldsymbol{Q}\vec{\boldsymbol{X}}_i[p] + \vec{\boldsymbol{t}}) + \sum_{j \in \mathcal{N}(i)} \beta_{ij} \left( \boldsymbol{\alpha}_{ij}[p] \cdot \left( \begin{bmatrix} \boldsymbol{Q}\vec{\boldsymbol{X}}_i[p] + \vec{\boldsymbol{t}} - (\boldsymbol{Q}\vec{\boldsymbol{X}}_j[1] + \vec{\boldsymbol{t}}) \\ \vdots \\ \boldsymbol{Q}\vec{\boldsymbol{X}}_i[p] + \vec{\boldsymbol{t}} - (\boldsymbol{Q}\vec{\boldsymbol{X}}_j[n_j] + \vec{\boldsymbol{t}}) \end{bmatrix} \odot \sigma_v(\boldsymbol{V}_j \parallel \text{RBF}(\boldsymbol{D}_{ij}[p])) \right) \odot \sigma_m(\boldsymbol{m}_{ij,p}) \right)$$

$$= g \cdot \vec{\boldsymbol{X}}_i[p] + \sum_{j \in \mathcal{N}(i)} \beta_{ij} \left( \boldsymbol{\alpha}_{ij}[p] \cdot \left( \begin{bmatrix} g \cdot \vec{\boldsymbol{X}}_i[p] - g \cdot \vec{\boldsymbol{X}}_j[1] \\ \vdots \\ g \cdot \vec{\boldsymbol{X}}_i[p] - g \cdot \vec{\boldsymbol{X}}_j[n_j] \end{bmatrix} \odot \sigma_v(\boldsymbol{V}_j \parallel \text{RBF}(\boldsymbol{D}_{ij}[p])) \right) \odot \sigma_m(\boldsymbol{m}_{ij,p}) \right),$$

which concludes the proof of Lemma C.3. $\qquad\square$

**Lemma C.4.** *Denote the equivariant feed-forward network as as $\{\boldsymbol{H}_i', \vec{\boldsymbol{X}}_i'\} = \mathrm{EFFN}(\{\boldsymbol{H}_i, \vec{\boldsymbol{X}}_i\})$, then it is E(3)-equivariant. Namely, $\forall g \in E(3)$, we have $\{\boldsymbol{H}_i', g \cdot \vec{\boldsymbol{X}}_i'\} = \mathrm{EFFN}(\{\boldsymbol{H}_i, g \cdot \vec{\boldsymbol{X}}_i\})$.*

*Proof.* The proof of Lemma C.4 focuses on the single-atom updates in Eq. 11-14. First, it is easy to obtain the E(3)-equivariance of the centroid in Eq. 11:

$$g \cdot \vec{\boldsymbol{x}}_c = g \cdot \mathrm{centroid}(\vec{\boldsymbol{X}}_i) = \mathrm{centroid}(g \cdot \vec{\boldsymbol{X}}_i).$$

Then we have the following equation on the relative coordinate $\Delta\vec{\boldsymbol{x}}$ in Eq. 12:

$$\boldsymbol{Q}\Delta\vec{\boldsymbol{x}} = (\boldsymbol{Q}\vec{\boldsymbol{x}} + \vec{\boldsymbol{t}}) - (\boldsymbol{Q}\vec{\boldsymbol{x}}_c + \vec{\boldsymbol{t}}) = g \cdot \vec{\boldsymbol{x}} - g \cdot \vec{\boldsymbol{x}}_c.$$

We can immediately obtain the E(3)-invariance of $\boldsymbol{r}$ in Eq. 12:

$$\boldsymbol{r} = \mathrm{RBF}(\|\boldsymbol{Q}\Delta\vec{\boldsymbol{x}}\|_2) = \mathrm{RBF}(\sqrt{(\boldsymbol{Q}\Delta\vec{\boldsymbol{x}})^\top(\boldsymbol{Q}\Delta\vec{\boldsymbol{x}})}) = \mathrm{RBF}(\sqrt{\Delta\vec{\boldsymbol{x}}^\top \boldsymbol{Q}^\top \boldsymbol{Q}\Delta\vec{\boldsymbol{x}}}) = \mathrm{RBF}(\|\Delta\vec{\boldsymbol{x}}\|_2).$$

Finally we can derive the E(3)-invariance on $\boldsymbol{h}$ and the E(3)-equivariance on $\vec{\boldsymbol{x}}$:

$$
\begin{aligned}
\boldsymbol{h}' &= \boldsymbol{h} + \phi_h(\boldsymbol{h}, \boldsymbol{h}_c, \boldsymbol{r}), \\
g \cdot \vec{\boldsymbol{x}}' &= g \cdot (\vec{\boldsymbol{x}} + \Delta\vec{\boldsymbol{x}}\phi_x(\boldsymbol{h}, \boldsymbol{h}_c, \boldsymbol{r})) \\
&= \boldsymbol{Q}(\vec{\boldsymbol{x}} + \Delta\vec{\boldsymbol{x}}\phi_x(\boldsymbol{h}, \boldsymbol{h}_c, \boldsymbol{r})) + \vec{\boldsymbol{t}} \\
&= \boldsymbol{Q}\vec{\boldsymbol{x}} + \vec{\boldsymbol{t}} + \boldsymbol{Q}\Delta\vec{\boldsymbol{x}}\phi_x(\boldsymbol{h}, \boldsymbol{h}_c, \boldsymbol{r}) \\
&= g \cdot \vec{\boldsymbol{x}} + (g \cdot \vec{\boldsymbol{x}} - g \cdot \vec{\boldsymbol{x}}_c)\phi_x(\boldsymbol{h}, \boldsymbol{h}_c, \boldsymbol{r}) \\
&= g \cdot \vec{\boldsymbol{x}} + (g \cdot \vec{\boldsymbol{x}} - \mathrm{centroid}(g \cdot \vec{\boldsymbol{X}}_i))\phi_x(\boldsymbol{h}, \boldsymbol{h}_c, \boldsymbol{r}),
\end{aligned}
$$

which concludes the proof of Lemma C.4 $\qquad\square$

**Lemma C.5.** *Denote the equivariant layer normalization as as $\{\boldsymbol{H}_i', \vec{\boldsymbol{X}}_i'\} = \mathrm{ELN}(\{\boldsymbol{H}_i, \vec{\boldsymbol{X}}_i\})$, then it is E(3)-equivariant. Namely, $\forall g \in E(3)$, we have $\{\boldsymbol{H}_i', g \cdot \vec{\boldsymbol{X}}_i'\} = \mathrm{ELN}(\{\boldsymbol{H}_i, g \cdot \vec{\boldsymbol{X}}_i\})$.*

*Proof.* Since the layer normalization is implemented on the atom level, namely each row of the coordinate matrix in a node, we again only need to concentrate on the single-atom normalization in Eq. 15-16. The key points lie in the E(3)-equivariance of $\mathbb{E}[\vec{\mathbf{X}}]$ and the E(3)-invariance of $\mathrm{Var}[\vec{\mathbf{X}} - \mathbb{E}[\vec{\mathbf{X}}]]$. The first one is obvious because $\mathbb{E}[\vec{\mathbf{X}}]$ is the centroid of the coordinates of all atoms:

$$g \cdot \mathbb{E}[\vec{\mathbf{X}}] = g \cdot \mathrm{centroid}(\vec{\mathbf{X}}) = \mathrm{centroid}(g \cdot \vec{\mathbf{X}}) = \mathbb{E}[g \cdot \vec{\mathbf{X}}].$$

Suppose there are $N$ atoms in total, then we can prove the E(3)-invariance of the variance as follows:

$$
\begin{aligned}
\mathrm{Var}[\vec{\mathbf{X}} - \mathbb{E}[\vec{\mathbf{X}}]] &= \frac{\sum_{i=1}^N (x_i - \bar{x})^2 + \sum_{i=1}^N (y_i - \bar{y})^2 + \sum_{i=1}^N (z_i - \bar{z})^2}{3N} \\
&= \frac{\sum_{i=1}^N [(x_i - \bar{x})^2 + (y_i - \bar{y})^2 + (z_i - \bar{z})^2]}{3N} \\
&= \frac{\sum_{i=1}^N (\vec{\boldsymbol{x}}_i - \mathbb{E}[\vec{\mathbf{X}}])^\top (\vec{\boldsymbol{x}}_i - \mathbb{E}[\vec{\mathbf{X}}])}{3N} \\
&= \frac{\sum_{i=1}^N (\vec{\boldsymbol{x}}_i - \mathbb{E}[\vec{\mathbf{X}}])^\top \boldsymbol{Q}^\top \boldsymbol{Q} (\vec{\boldsymbol{x}}_i - \mathbb{E}[\vec{\mathbf{X}}])}{3N} \\
&= \frac{\sum_{i=1}^N (\boldsymbol{Q}\vec{\boldsymbol{x}}_i - \boldsymbol{Q}\mathbb{E}[\vec{\mathbf{X}}])^\top (\boldsymbol{Q}\vec{\boldsymbol{x}}_i - \boldsymbol{Q}\mathbb{E}[\vec{\mathbf{X}}])}{3N} \\
&= \frac{\sum_{i=1}^N (g \cdot \vec{\boldsymbol{x}}_i - g \cdot \mathbb{E}[\vec{\mathbf{X}}])^\top (g \cdot \vec{\boldsymbol{x}}_i - g \cdot \mathbb{E}[\vec{\mathbf{X}}])}{3N} \\
&= \frac{\sum_{i=1}^N (g \cdot \vec{\boldsymbol{x}}_i - \mathbb{E}[g \cdot \vec{\mathbf{X}}])^\top (g \cdot \vec{\boldsymbol{x}}_i - \mathbb{E}[g \cdot \vec{\mathbf{X}}])}{3N} \\
&= \mathrm{Var}[g \cdot \vec{\mathbf{X}} - \mathbb{E}[g \cdot \vec{\mathbf{X}}]].
\end{aligned}
$$

Therefore, we can finally derive the E(3)-invariance on $h$ and the E(3)-equivariance on $\vec{x}$ in Eq. 15-16:

$$h = \frac{h - \mathbb{E}[h]}{\sqrt{\mathrm{Var}[h]}} \cdot \gamma + \beta,$$

$$g \cdot \vec{x} = g \cdot (\frac{\vec{x} - \mathbb{E}[\vec{\mathbf{X}}]}{\sqrt{\mathrm{Var}[\vec{\mathbf{X}} - \mathbb{E}[\vec{\mathbf{X}}]]}} \cdot \sigma + \mathbb{E}[\vec{\mathbf{X}}]) = \frac{Q\vec{x} - Q\mathbb{E}[\vec{\mathbf{X}}]}{\sqrt{\mathrm{Var}[\vec{\mathbf{X}} - \mathbb{E}[\vec{\mathbf{X}}]]}} \cdot \sigma + Q\mathbb{E}[\vec{\mathbf{X}}] + \vec{t}$$

$$= \frac{Q\vec{x} + \vec{t} - (Q\mathbb{E}[\vec{\mathbf{X}}] + \vec{t})}{\sqrt{\mathrm{Var}[\vec{\mathbf{X}} - \mathbb{E}[\vec{\mathbf{X}}]]}} \cdot \sigma + (Q\mathbb{E}[\vec{\mathbf{X}}] + \vec{t}) = \frac{g \cdot \vec{x} - g \cdot \mathbb{E}[\vec{\mathbf{X}}]}{\sqrt{\mathrm{Var}[g \cdot \vec{\mathbf{X}} - \mathbb{E}[g \cdot \vec{\mathbf{X}}]]}} \cdot \sigma + g \cdot \mathbb{E}[\vec{\mathbf{X}}]$$

$$= \frac{g \cdot \vec{x} - \mathbb{E}[g \cdot \vec{\mathbf{X}}]}{\sqrt{\mathrm{Var}[g \cdot \vec{\mathbf{X}} - \mathbb{E}[g \cdot \vec{\mathbf{X}}]]}} \cdot \sigma + \mathbb{E}[g \cdot \vec{\mathbf{X}}],$$

which concludes the proof of Lemma C.5. $\qquad\square$

With Lemma C.3-C.5 at hand, it is obvious to deduce the E(3)-equivariance of the GET layer.

## C.2 PROOF OF INTRA-BLOCK PERMUATION INVARIANCE

Obviously, the feed-forward network and the layer normalization are invariant to intra-block permutations because they are implemented on single atoms and the only incorporated multi-atom operation is averaging, which is invariant to the permutations. Therefore, the proof narrows down to the intra-block permutation invariance of the bilevel attention module.

**Lemma C.6.** *Denote the bilevel attention module as* $\{H_i', \vec{X}_i'\} = \mathrm{Att}(\{H_i, \vec{X}_i\})$, *then it conforms to intra-block permutation invariance. Namely,* $\forall\{\pi_i \in S_{n_i} | 1 \le i \le B\}$, *where* $B$ *is the number of blocks in the input and* $S_{n_i}$ *includes all permutations on* $n_i$ *elements, we have* $\{\pi_i \cdot H_i', \pi_i \cdot \vec{X}_i'\} = \mathrm{Att}(\{\pi_i \cdot H_i, \pi_i \cdot \vec{X}_i\})$.

*Proof.* Denote the the permutation of block $i$ as $\pi_i$, then it can be instantiated as the multiplication of a series of elementary row-switching matrices $P_i = P_i^{(m_i)} P_i^{(m_i-1)} \dots P_i^{(1)}$. For example, we have $\pi_i \cdot H_i = P_i H_i$, $\pi_i \cdot \vec{X}_i = P_i \vec{X}_i$. Here we first prove an elegant property of $P_i$, which we will use in the later proof:

$$\begin{aligned}
P_i^\top P_i &= (P_i^{(m_i)} P_i^{(m_i-1)} \dots P_i^{(1)})^\top (P_i^{(m_i)} P_i^{(m_i-1)} \dots P_i^{(1)}) \\
&= P_i^{(1)^\top} \dots P_i^{(m_i-1)^\top} P_i^{(m_i)^\top} P_i^{(m_i)} P_i^{(m_i-1)} \dots P_i^{(1)} \\
&= P_i^{(1)^\top} \dots P_i^{(m_i-1)^\top} I P_i^{(m_i-1)} \dots P_i^{(1)} \\
&= \dots \\
&= I
\end{aligned}$$

Given arbitrary permutations on each block, we have the permutated query, key, and value matrics:

$$\begin{aligned}
P_i Q_i &= P_i H_i W_Q = (\pi_i \cdot H_i) W_Q, \\
P_i K_i &= P_i H_i W_K = (\pi_i \cdot H_i) W_K, \\
P_i V_i &= P_i H_i W_V = (\pi_i \cdot H_i) W_V.
\end{aligned}$$

The distance matrix $\boldsymbol{D}_{ij}$ is also permutated as $\boldsymbol{P}_i \boldsymbol{D}_{ij} \boldsymbol{P}_j^\top$. Therefore, the atom-level attention $\boldsymbol{\alpha}_{ij}$ in Eq. 4 is also permutated as $\boldsymbol{P}_i \boldsymbol{\alpha}_{ij} \boldsymbol{P}_j^\top$, and the messages in Eq. 7-8 are permutated as:

$$\boldsymbol{P}_i \boldsymbol{m}_{ij} = \boldsymbol{P}_i \begin{bmatrix} \boldsymbol{\alpha}_{ij}[1] \cdot \phi_v(\boldsymbol{V}_j \| \mathrm{RBF}(\boldsymbol{D}_{ij}[1])) \\ \vdots \\ \boldsymbol{\alpha}_{ij}[n_i] \cdot \phi_v(\boldsymbol{V}_j \| \mathrm{RBF}(\boldsymbol{D}_{ij}[n_i])) \end{bmatrix} = \boldsymbol{P}_i \begin{bmatrix} \boldsymbol{\alpha}_{ij}[1] \boldsymbol{P}_j^\top \boldsymbol{P}_j \phi_v(\boldsymbol{V}_j \| \mathrm{RBF}(\boldsymbol{D}_{ij}[1])) \\ \vdots \\ \boldsymbol{\alpha}_{ij}[n_i] \boldsymbol{P}_j^\top \boldsymbol{P}_j \phi_v(\boldsymbol{V}_j \| \mathrm{RBF}(\boldsymbol{D}_{ij}[n_i])) \end{bmatrix}$$

$$\boldsymbol{P}_i \vec{\boldsymbol{m}}_{ij} = \boldsymbol{P}_i \begin{bmatrix} \boldsymbol{\alpha}_{ij}[1] \cdot \left( \vec{\boldsymbol{X}}_{ij}[p] \odot \sigma_v(\boldsymbol{V}_j \| \mathrm{RBF}(\boldsymbol{D}_{ij}[1])) \right) \\ \vdots \\ \boldsymbol{\alpha}_{ij}[n_i] \cdot \left( \vec{\boldsymbol{X}}_{ij}[p] \odot \sigma_v(\boldsymbol{V}_j \| \mathrm{RBF}(\boldsymbol{D}_{ij}[n_i])) \right) \end{bmatrix}$$

$$= \boldsymbol{P}_i \begin{bmatrix} \boldsymbol{\alpha}_{ij}[1] \boldsymbol{P}_j^\top \boldsymbol{P}_j \left( \vec{\boldsymbol{X}}_{ij}[p] \odot \sigma_v(\boldsymbol{V}_j \| \mathrm{RBF}(\boldsymbol{D}_{ij}[1])) \right) \\ \vdots \\ \boldsymbol{\alpha}_{ij}[n_i] \boldsymbol{P}_j^\top \boldsymbol{P}_j \left( \vec{\boldsymbol{X}}_{ij}[p] \odot \sigma_v(\boldsymbol{V}_j \| \mathrm{RBF}(\boldsymbol{D}_{ij}[n_i])) \right) \end{bmatrix},$$

The block-level attention $\beta_{ij}$ in Eq. 6 remains unchanged as the average of $\boldsymbol{R}_{ij}$ in obtaining $\boldsymbol{r}_{ij}$ eliminates the effect of permutations. Finally, we can derive the intra-block permutation invariance as follows:

$$\boldsymbol{P}_i \boldsymbol{H}_i' = \boldsymbol{P}_i \left( \boldsymbol{H}_i + \sum_{j \in \mathcal{N}(i)} \beta_{ij} \phi_m(\boldsymbol{m}_{ij}) \right) = \boldsymbol{P}_i \boldsymbol{H}_i + \sum_{j \in \mathcal{N}(i)} \beta_{ij} \phi_m(\boldsymbol{P}_i \boldsymbol{m}_{ij}),$$

$$\boldsymbol{P}_i \vec{\boldsymbol{X}}_i' = \boldsymbol{P}_i \left( \vec{\boldsymbol{X}}_i + \sum_{j \in \mathcal{N}(i)} \beta_{ij} \vec{\boldsymbol{m}}_{ij} \odot \sigma_m(\vec{\boldsymbol{m}}_{ij}) \right) = \boldsymbol{P}_i \vec{\boldsymbol{X}}_i + \sum_{j \in \mathcal{N}(i)} \beta_{ij} (\boldsymbol{P}_i \vec{\boldsymbol{m}}_{ij}) \odot \sigma_m(\boldsymbol{P}_i \vec{\boldsymbol{m}}_{ij})$$

which concludes Lemma C.6. $\qquad\square$

## D   COMPLEXITY ANALYSIS

To discuss the scalability of the model, we additionally provide the complexity analysis as follows. The main complexity lies in the attention-based message passing module. Suppose block $i$ and block $j$ have $n_i$ and $n_j$ atoms, respectively. Since the attention module implements bipartite cross attention between block pairs, there are a total of $n_i n_j$ attention edges between block $i$ and block $j$. Therefore, the exact complexity should be $O(\sum_{i \in \mathcal{V}} \sum_{j \in \mathcal{N}(i)} n_i n_j)$, where $\mathcal{V}$ includes all nodes and $\mathcal{N}(i)$ includes all neighbors of block $i$. Since we use $K$ nearest neighbors to construct graphs in block level, we have $\mathcal{N}(i) \le K$. Denote the maximum number of atoms in a single block is $C$(in natural proteins we have $C = 14$), we have $n_i \le C$. Therefore, we have $\sum_{i \in \mathcal{V}} \sum_{j \in \mathcal{N}(i)} n_i n_j \le \sum_{i \in \mathcal{V}} KC^2 = KC^2 |\mathcal{V}|$, namely, the complexity should be bounded by $O(KC^2 |\mathcal{V}|)$, which is linear to the number of blocks in the graph. A linear complexity means the algorithm should be easy to scale to larger molecular systems.

Practically, the complexity can be further optimized by selecting only $k$ nearest neighbors of each atom in message passing between block $i$ and block $j$. With this sparse attention, the complexity is $O(\sum_{i \in \mathcal{V}} \sum_{j \in \mathcal{N}(i)} k n_i) \le O(kKN)$, where $N$ is the total number of atoms.

## E   LIGAND EFFICACY PREDICTION

We additionally provide the evaluation results on Ligand Efficacy Prediction (LEP). This task requires identifying a given ligand as the "activator" or the "inactivator" of a functional protein. Specifically, given the two complexes where the ligand interacts with the active and the inactive conformation of the protein respectively, the models need to distinguish which one is more favorable. To this end, we first obtain the graph-level representations of the two complexes. Then we concatenate the two representations to do a binary classification. We use two metrics for evaluation: the area under the receiver operating characteristic (**AUROC**) and the area under precision-recall curve (**AUPRC**).

**Dataset** We follow the LEP dataset and its splits in the Atom3D benchmark (Townshend et al., 2020), which includes 27 functional proteins and 527 ligands known as activator or inactivator to a certain protein. The active and the inactive complexes are generated by Glide (Friesner et al., 2004). The splits of the training, the validation, and the test sets are based on the functional proteins to ensure generalizability.

Table 5: The mean and the standard deviation of three runs on ligand efficacy prediction. The best results are marked in bold and the second best are underlined.

| Repr. | Model | AUROC↑ | AUPRC↑ |
|---|---|---|---|
| Block | SchNet | $0.732 \pm 0.022$ | $0.718 \pm 0.031$ |
| | DimeNet++ | $0.669 \pm 0.014$ | $0.609 \pm 0.036$ |
| | EGNN | $\underline{0.746 \pm 0.017}$ | $\mathbf{0.755 \pm 0.031}$ |
| | ET | $0.744 \pm 0.034$ | $0.721 \pm 0.052$ |
| Atom | SchNet | $0.712 \pm 0.026$ | $0.639 \pm 0.033$ |
| | DimeNet++ | $0.589 \pm 0.049$ | $0.503 \pm 0.020$ |
| | EGNN | $0.711 \pm 0.020$ | $0.643 \pm 0.041$ |
| | ET | $0.677 \pm 0.004$ | $0.636 \pm 0.054$ |
| Hierarchical | SchNet | $0.736 \pm 0.020$ | $0.731 \pm 0.048$ |
| | DimeNet++ | $0.579 \pm 0.118$ | $0.517 \pm 0.100$ |
| | EGNN | $0.724 \pm 0.027$ | $0.720 \pm 0.056$ |
| | ET | $0.717 \pm 0.033$ | $0.724 \pm 0.055$ |
| Unified | GET (ours) | $\mathbf{0.761 \pm 0.012}$ | $\underline{0.751 \pm 0.012}$ |

**Results** We present the mean and the standard deviation of the metrics across three runs in Table 5. LEP requires distinguishing the active and inactive conformations of the receptor, thus it is essential to capture the block-level geometry of the protein in addition to the atom-level receptor-ligand interactions. The unified representation excels at learning the bilevel geometry, therefore, naturally, we observe obvious gains on the metrics of our method compared to the baselines.

## F IMPLEMENTATION DETAILS

We conduct experiments on 1 GeForce RTX 2080 Ti GPU. Each model is trained with Adam optimizer and exponential learning rate decay. To avoid unstable checkpoints from early training stages, we select the latest checkpoint from the saved top-$k$ checkpoints on the validation set for testing. Since the number of blocks varies a lot in different samples, we set a upperbound of the number of blocks to form a dynamic batch instead of using a static batch size. We use $k = 9$ for constructing the k-nearest neighbor graph in § 3.1 and set the size of the RBF kernel ($d_{\mathrm{rbf}}$) to 32. We give the description of the hyperparameters in Table 6 and their values for each task in Table 7.

Table 6: Descriptions of the hyperparameters.

| hyperparameter | description |
|---|---|
| $d_h$ | Hidden size |
| $d_r$ | Radial size for the attention module |
| lr | Learning rate |
| final_lr | Final learning rate |
| max_epoch | Maximum of epochs to train |
| save_topk | Number of top-$k$ checkpoints to save |
| n_layers | Number of layers |
| max_n_vertex | Upperbound of the number of nodes in a batch |

### F.1 PDBBIND BENCHMARK

We follow Somnath et al. (2021); Wang et al. (2022) to conduct experiments on the well-established PDBbind (Wang et al., 2004; Liu et al., 2015) and use the split with sequence identity threshold of 30% which should barely have data leakage problem. A total of 4709 complexes are first filtered by resolution and then splitted into 3507, 466, 490 for training, validation, and testing (Somnath et al., 2021). We directly borrow the results of the baselines from Wang et al. (2022). For our model, we set $d_h = 64, d_r = 64, \mathrm{lr} = 10^{-3}, \mathrm{final\_lr} = 10^{-4}$, and max_n_vertex $= 1500$.

Table 7: Hyperparameters for our GET on each task.

| hyperparameter | PPA | LBA | LEP | hyperparameter | PPA | LBA | LEP |
|---|---|---|---|---|---|---|---|
| GET | | | | | | | |
| $d_h$ | 128 | 64 | 128 | $d_r$ | 16 | 32 | 64 |
| lr | $10^{-4}$ | $10^{-3}$ | $5 \times 10^{-4}$ | final_lr | $10^{-4}$ | $10^{-6}$ | $10^{-4}$ |
| max_epoch | 20 | 10 | 90 | save_topk | 3 | 3 | 7 |
| n_layers | 3 | 3 | 3 | max_n_vertex | 1500 | 2000 | 1500 |
| GET-mix | | | | | | | |
| $d_h$ | 128 | 128 | - | $d_r$ | 16 | 16 | - |
| lr | $5 \times 10^{-5}$ | $5 \times 10^{-5}$ | - | final_lr | $5 \times 10^{-5}$ | $10^{-6}$ | - |
| max_epoch | 20 | 20 | - | save_topk | 3 | 3 | - |
| n_layers | 3 | 3 | - | max_n_vertex | 1500 | 1500 | - |

### F.2 PROTEIN-PROTEIN AFFINITY

Here we illustrate the setup for protein-protein affinity prediction with more details.

We adopt the Protein-Protein Affinity Benchmark Version 2 (Kastritis et al., 2011; Vreven et al., 2015) as the test set, which contains 176 diversified protein-protein complexes with annotated affinity collected from existing literature. These complexes are further categorized into three difficulty levels (i.e. Rigid, Medium, Flexible) according to the conformation change of the proteins from the unbound to the bound state (Kastritis et al., 2011), among which the Flexible split is the most challenging as the proteins undergo large conformation change upon binding.

As for training, we first filter out 2,500 complexes with annotated binding affinity ($K_i$ or $K_d$) from PDBbind (Wang et al., 2004). Then we use MMseqs2 (Steinegger & Söding, 2017) to cluster the sequences of these complexes together with the test set by dividing complexes with sequence identity above 30% into the same cluster, where sequence identity is calculated based on the BLOSUM62 substitution matrix (Henikoff & Henikoff, 1992). The complexes that shares the same clusters with the test set are dropped to prevent data leakage, after which we finally obtained 2,195 valid complexes. We split these complexes into training set and validation set with a ratio of 9:1 with respect to the number of clusters. Following previous literature (Ballester & Mitchell, 2010; Jiménez et al., 2018; Ragoza et al., 2017), we predict the negative log-transformed value ($pK$) instead of direct regression on the affinity.

## G BASELINES

### G.1 IMPLEMENTATION

In this section, we describe the implementation details of different baselines. All the baselines are designed for structural learning on graphs whose nodes are represented as one feature vector and one coordinate. Therefore, for **block-level** representation, we average the embeddings and the coordinates of the atoms in each block before feeding the graph to the baselines. For **atom-level** representation, each node is represented as the embedding and the coordinate of each atom. For **Hierarchical** methods, we first implement message passing on atom-level graphs, then average the embeddings and coordinates within each block before conducting block-level message passing (Jin et al., 2022). For fair comparison, the number of layers in each model is set to 3, except MACE and Equiformer, which are quite unstable in training with more than 2 layers. We present other hyperparameters in Table 8.

For **SchNet** (Schütt et al., 2017) and **DimeNet++** (Gasteiger et al., 2020b;a), we use the implementation in PyTorch Geometric (Fey & Lenssen, 2019). For **EGNN** (Satorras et al., 2021), **ET** (Thölke & De Fabritiis, 2022), **MACE** (Batatia et al., 2022), and **LEFTNet** (Du et al., 2023), we directly use the official open-source codes provided in their papers. For **GemNet** (Gasteiger et al., 2021) and **Equiformer** (Liao & Smidt, 2022), we use the implementation in open-source projects, the Open Catalyst Project (Chanussot* et al., 2021) and equiformer-pytorch[2], respectively. For fair comparison with SchNet and ET, we project the edge feature into the same shape as the distance feature in these models, and add the edge feature to the distance feature. It is also worth mentioning that due

---

[2] https://github.com/lucidrains/equiformer-pytorch/tree/main

to the high complexity of angular features (DimeNet++, GemNet) and irreducible representations (MACE, Equiformer), these models need 2 GeForce RTX 2080 Ti GPUs for training on atomic graphs. Even so, some of them still fail to run the experiments due to the limitation of the GPU memory (*i.e.* GemNet and Equiformer).

Table 8: Hyperparameters for each baseline on each task.

| hyperparameter | PPA | LBA | LEP | hyperparameter | PPA | LBA | LEP |
|---|---|---|---|---|---|---|---|
| | | | SchNet | | | | |
| $d_h$ | 128 | 64 | 64 | max_n_vertex | 1500 | 1500 | 1500 |
| lr | $10^{-3}$ | $5 \times 10^{-4}$ | $10^{-3}$ | final_lr | $10^{-4}$ | $10^{-5}$ | $10^{-4}$ |
| max_epoch | 20 | 60 | 65 | save_topk | 3 | 5 | 5 |
| | | | SchNet-mix | | | | |
| $d_h$ | 128 | 128 | - | max_n_vertex | 1500 | 1500 | - |
| lr | $5 \times 10^{-5}$ | $5 \times 10^{-5}$ | - | final_lr | $5 \times 10^{-5}$ | $5 \times 10^{-5}$ | - |
| max_epoch | 20 | 20 | - | save_topk | 3 | 3 | - |
| | | | DimeNet++ | | | | |
| $d_h$ | 128 | 64 | 64 | max_n_vertex | 1500 | 1500 | 1500 |
| lr | $10^{-3}$ | $5 \times 10^{-4}$ | $10^{-3}$ | final_lr | $10^{-4}$ | $10^{-5}$ | $10^{-4}$ |
| max_epoch | 20 | 60 | 65 | save_topk | 3 | 5 | 5 |
| | | | EGNN | | | | |
| $d_h$ | 128 | 64 | 64 | max_n_vertex | 1500 | 1500 | 1500 |
| lr | $10^{-3}$ | $5 \times 10^{-4}$ | $10^{-3}$ | final_lr | $10^{-4}$ | $10^{-5}$ | $10^{-4}$ |
| max_epoch | 20 | 60 | 65 | save_topk | 3 | 5 | 5 |
| | | | ET | | | | |
| $d_h$ | 128 | 64 | 64 | max_n_vertex | 1500 | 1500 | 1500 |
| lr | $10^{-3}$ | $5 \times 10^{-4}$ | $10^{-3}$ | final_lr | $10^{-4}$ | $10^{-5}$ | $10^{-4}$ |
| max_epoch | 20 | 20 | 65 | save_topk | 3 | 3 | 5 |
| | | | ET-mix | | | | |
| $d_h$ | 128 | 128 | - | max_n_vertex | 1500 | 1500 | - |
| lr | $5 \times 10^{-5}$ | $5 \times 10^{-5}$ | - | final_lr | $5 \times 10^{-5}$ | $5 \times 10^{-5}$ | - |
| max_epoch | 20 | 20 | - | save_topk | 3 | 3 | - |
| | | | GemNet | | | | |
| $d_h$ | 128 | 64 | - | max_n_vertex | 1000 | 2000 | - |
| lr | $10^{-4}$ | $10^{-3}$ | - | final_lr | $10^{-4}$ | $10^{-6}$ | - |
| max_epoch | 20 | 10 | - | save_topk | 3 | 3 | - |
| | | | MACE | | | | |
| $d_h$ | 128 | 64 | - | max_n_vertex | 1500 | 1500 | - |
| lr | $10^{-4}$ | $10^{-3}$ | - | final_lr | $10^{-4}$ | $10^{-6}$ | - |
| max_epoch | 20 | 20 | - | save_topk | 3 | 3 | - |
| | | | MACE-mix | | | | |
| $d_h$ | 128 | 128 | - | max_n_vertex | 1500 | 1500 | - |
| lr | $5 \times 10^{-5}$ | $5 \times 10^{-5}$ | - | final_lr | $5 \times 10^{-5}$ | $5 \times 10^{-5}$ | - |
| max_epoch | 20 | 20 | - | save_topk | 3 | 3 | - |
| | | | Equiformer[3] | | | | |
| $d_h$ | 32 | 32 | - | max_n_vertex | 400 | 1000 | - |
| lr | $10^{-4}$ | $10^{-3}$ | - | final_lr | $10^{-4}$ | $10^{-6}$ | - |
| max_epoch | 10 | 10 | - | save_topk | 3 | 3 | - |
| | | | LEFTNet | | | | |
| $d_h$ | 128 | 64 | - | max_n_vertex | 1000 | 2000 | - |
| lr | $10^{-4}$ | $10^{-3}$ | - | final_lr | $10^{-4}$ | $10^{-6}$ | - |
| max_epoch | 20 | 10 | - | save_topk | 3 | 3 | - |
| | | | LEFTNet-mix | | | | |
| $d_h$ | 128 | 128 | - | max_n_vertex | 1000 | 1000 | - |
| lr | $10^{-4}$ | $10^{-4}$ | - | final_lr | $10^{-4}$ | $10^{-4}$ | - |
| max_epoch | 20 | 20 | - | save_topk | 3 | 3 | - |

### G.2 NUMBER OF PARAMETERS AND TRAINING EFFICIENCY

We further provide the number of parameters and training efficiency for the baselines as well as our GET in Table 9.

When comparing GET to simpler yet weaker baselines (e.g., SchNet and EGNN), it is evident that GET may have more parameters and a slower training speed. However, it is crucial to note that GET exhibits competitive parameter and computation efficiency when compared with more complex yet stronger baselines, such as Equiformer, MACE, and LEFTNet. It's worth mentioning that a significant portion of parameters in GET is attributed to the Feedforward Neural Network (FFN) that projects latent features to higher dimensions in intermediate layers, aligning with the structure of vanilla Transformers. Without this part, GET has the least parameters among all the models.

---

[3] Equiformer is quite unstable and extremely memory intensive with large widths and depths.

Nevertheless, adding FFN should not harm the efficiency much as the time cost mainly comes from message passing over edges, which is propotional to the number of edges, while time complexity of FFN is propotional to the number of nodes whose value is much smaller than the number of edges.

Moreover, the throughput of GET is comparable to both atom-level and hierarchical counterparts. This aligns with the design of the model, which considers both block-level and atom-level geometry. We also present the complexity analysis in Appendix D to further elucidate the efficiency of our approach, showing its linear complexity concerning the number of nodes in the graph. This characteristic indicates favorable scalability to large graphs in practical settings.

Table 9: Number of paramters and training speed for baselines and our GET.

| Repr. | Model | PPA | | LBA | | LEP | |
|---|---|---|---|---|---|---|---|
| | | Parameter | Sec. / Batch | Parameter | Sec. / Batch | Parameter | Sec. / Batch |
| Block | SchNet | 0.25M | 0.054 | 0.15M | 0.040 | 0.14M | 0.118 |
| | DimeNet++ | 1.53M | 0.233 | 0.40M | 0.189 | 0.40M | 0.263 |
| | EGNN | 0.43M | 0.054 | 0.12M | 0.035 | 0.11M | 0.130 |
| | ET | 0.71M | 0.072 | 0.20M | 0.050 | 0.20M | 0.133 |
| | GemNet | 1.35M | 0.225 | 0.69M | 0.179 | - | - |
| | MACE | 12.9M | 0.296 | 1.97M | 0.285 | - | - |
| | Equiformer | 0.56M | 1.846 | 0.56M | 1.364 | - | - |
| | LEFTNet | 1.57M | 0.088 | 0.43M | 0.068 | - | - |
| Atom | SchNet | 0.25M | 0.109 | 0.15M | 0.050 | 0.14M | 0.123 |
| | DimeNet++ | 1.56M | OOM | 0.40M | 0.435 | 0.40M | 0.357 |
| | EGNN | 0.43M | 0.145 | 0.12M | 0.060 | 0.11M | 0.139 |
| | ET | 0.71M | 0.217 | 0.20M | 0.079 | 0.20M | 0.145 |
| | GemNet | 1.35M | OOM | 0.69M | OOM | - | - |
| | MACE | 12.9M | 1.259 | 1.97M | 0.535 | - | - |
| | Equiformer | 0.56M | OOM | 0.56M | OOM | - | - |
| | LEFTNet | 1.57M | 0.472 | 0.43M | 0.177 | - | - |
| Hierarchical | SchNet | 0.37M | 0.127 | 0.21M | 0.081 | 0.20M | 0.088 |
| | DimeNet++ | 3.07M | OOM | 0.60M | 0.622 | 0.60M | 0.633 |
| | EGNN | 0.61M | 0.143 | 0.17M | 0.077 | 0.17M | 0.100 |
| | ET | 1.00M | 0.184 | 0.30M | 0.104 | 0.29M | 0.119 |
| | GemNet | 2.64M | OOM | 1.37M | OOM | - | - |
| | MACE | 25.7M | 0.821 | 3.91M | 0.426 | - | - |
| | Equiformer | 1.10M | OOM | 1.10M | OOM | - | - |
| | LEFTNet | 3.10M | 0.307 | 0.85M | 0.129 | - | - |
| Unified | GET (w/o FFN) | 0.23M | 0.291 | 0.09M | 0.193 | 0.20M | 0.155 |
| | GET | 2.50M | 0.339 | 0.69M | 0.237 | 1.60M | 0.192 |

## H    SENSITIVITY TO WIDTH AND DEPTH

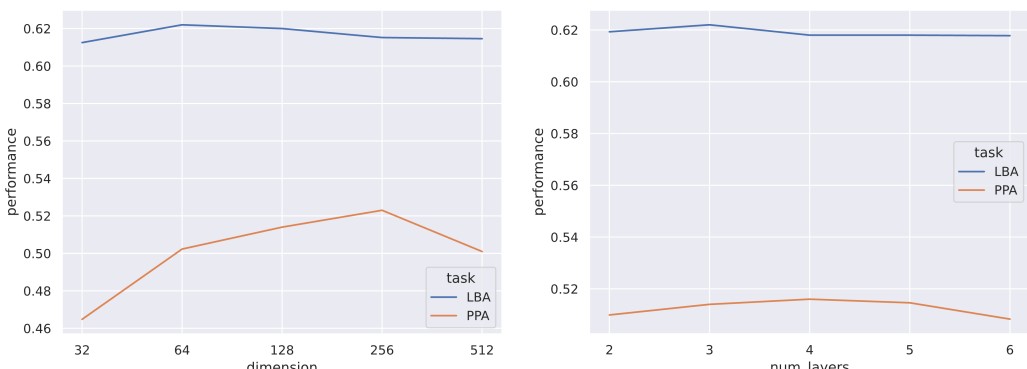

Figure 5: Performance with respect to the dimensions of the hidden layers (left) and the number of layers (right) on protein-protein affinity (PPA) and ligand-binding affinity (LBA).

We show the performance with respect to dimensions of the hidden layers and the number of layers in Figure 5 on protein-protein affinity (PPA) and ligand-binding affinity (LBA). The performance is not so sensitive to both width and depth on LBA, while it is relatively more sensitive to width on PPA. Nevertheless, the common trend is that performance increases first and then decreases with dimension getting larger or the network getting deeper, so it is still necessary to find the suitable size of the model that the dataset can support.

# I DETAILED RESULTS OF PROTEIN-PROTEIN AFFINITY

We show the detailed mean and standard deviation of three runs on all test splits of protein-protein affinity in Table 10.

Table 10: The mean and the standard deviation of three runs on protein-protein affinity prediction. The best results are marked in bold and the second best are underlined. Baselines that fail to process atomic graphs due to high complexity are marked with OOM (out of memory)

| Repr. | Model | Rigid | Medium | Flexible | All |
|---|---|---|---|---|---|
| | | Pearson↑ | | | |
| Block | SchNet | $0.542 \pm 0.012$ | $0.504 \pm 0.020$ | $0.102 \pm 0.019$ | $0.439 \pm 0.016$ |
| | DimeNet++ | $0.487 \pm 0.087$ | $0.367 \pm 0.043$ | $0.152 \pm 0.078$ | $0.323 \pm 0.025$ |
| | EGNN | $0.437 \pm 0.023$ | $0.436 \pm 0.028$ | $0.094 \pm 0.049$ | $0.381 \pm 0.021$ |
| | ET | $0.575 \pm 0.041$ | $0.470 \pm 0.024$ | $0.087 \pm 0.024$ | $0.424 \pm 0.021$ |
| | GemNet | $0.480 \pm 0.061$ | $0.425 \pm 0.051$ | $0.086 \pm 0.048$ | $0.387 \pm 0.023$ |
| | MACE | $0.621 \pm 0.022$ | $0.450 \pm 0.027$ | $0.307 \pm 0.041$ | $0.470 \pm 0.015$ |
| | Equiformer | $0.630 \pm 0.024$ | $0.503 \pm 0.015$ | $\underline{0.298 \pm 0.017}$ | $\underline{0.484 \pm 0.007}$ |
| | LEFTNet | $0.563 \pm 0.035$ | $0.497 \pm 0.018$ | $0.202 \pm 0.016$ | $0.452 \pm 0.013$ |
| Atom | SchNet | $0.592 \pm 0.007$ | $\mathbf{0.522 \pm 0.010}$ | $-0.038 \pm 0.016$ | $0.369 \pm 0.007$ |
| | DimeNet++ | | OOM | | |
| | EGNN | $0.497 \pm 0.027$ | $0.452 \pm 0.012$ | $-0.054 \pm 0.013$ | $0.302 \pm 0.010$ |
| | ET | $0.609 \pm 0.023$ | $0.486 \pm 0.004$ | $0.049 \pm 0.009$ | $0.401 \pm 0.005$ |
| | GemNet | | OOM | | |
| | MACE | $\underline{0.653 \pm 0.066}$ | $0.499 \pm 0.053$ | $0.241 \pm 0.061$ | $0.463 \pm 0.052$ |
| | Equiformer | | OOM | | |
| | LEFTNet | $0.583 \pm 0.080$ | $0.510 \pm 0.029$ | $0.243 \pm 0.091$ | $0.448 \pm 0.046$ |
| Hierarchical | SchNet | $0.542 \pm 0.028$ | $0.507 \pm 0.020$ | $0.098 \pm 0.011$ | $0.438 \pm 0.017$ |
| | DimeNet++ | | OOM | | |
| | EGNN | $0.461 \pm 0.018$ | $0.440 \pm 0.024$ | $0.089 \pm 0.051$ | $0.386 \pm 0.021$ |
| | ET | $0.572 \pm 0.051$ | $0.498 \pm 0.025$ | $0.101 \pm 0.093$ | $0.438 \pm 0.026$ |
| | GemNet | | OOM | | |
| | MACE | $0.616 \pm 0.069$ | $0.461 \pm 0.050$ | $0.275 \pm 0.032$ | $0.466 \pm 0.020$ |
| | Equiformer | | OOM | | |
| | LEFTNet | $0.533 \pm 0.059$ | $0.494 \pm 0.026$ | $0.165 \pm 0.031$ | $0.445 \pm 0.024$ |
| Unified | GET (ours) | $\mathbf{0.670 \pm 0.017}$ | $\underline{0.512 \pm 0.010}$ | $\mathbf{0.381 \pm 0.014}$ | $\mathbf{0.514 \pm 0.011}$ |
| | | Spearman↑ | | | |
| Block | SchNet | $0.476 \pm 0.015$ | $0.520 \pm 0.013$ | $0.068 \pm 0.009$ | $0.427 \pm 0.012$ |
| | DimeNet++ | $0.466 \pm 0.088$ | $0.368 \pm 0.037$ | $0.171 \pm 0.054$ | $0.317 \pm 0.031$ |
| | EGNN | $0.364 \pm 0.043$ | $0.455 \pm 0.026$ | $0.080 \pm 0.038$ | $0.382 \pm 0.022$ |
| | ET | $0.552 \pm 0.039$ | $0.482 \pm 0.025$ | $0.090 \pm 0.062$ | $0.415 \pm 0.027$ |
| | GemNet | $0.420 \pm 0.072$ | $0.446 \pm 0.059$ | $0.066 \pm 0.058$ | $0.393 \pm 0.027$ |
| | MACE | $0.596 \pm 0.047$ | $0.450 \pm 0.014$ | $\underline{0.306 \pm 0.029}$ | $0.466 \pm 0.011$ |
| | Equiformer | $0.560 \pm 0.015$ | $\underline{0.530 \pm 0.017}$ | $0.251 \pm 0.002$ | $\underline{0.496 \pm 0.007}$ |
| | LEFTNet | $0.515 \pm 0.039$ | $0.492 \pm 0.020$ | $0.193 \pm 0.023$ | $0.452 \pm 0.013$ |
| Atom | SchNet | $0.546 \pm 0.005$ | $0.512 \pm 0.007$ | $0.028 \pm 0.032$ | $0.404 \pm 0.016$ |
| | DimeNet++ | | OOM | | |
| | EGNN | $0.450 \pm 0.042$ | $0.438 \pm 0.021$ | $0.027 \pm 0.030$ | $0.349 \pm 0.009$ |
| | ET | $0.582 \pm 0.025$ | $0.487 \pm 0.002$ | $0.117 \pm 0.008$ | $0.436 \pm 0.004$ |
| | GemNet | | OOM | | |
| | MACE | $\underline{0.619 \pm 0.037}$ | $0.487 \pm 0.049$ | $0.221 \pm 0.064$ | $0.449 \pm 0.052$ |
| | Equiformer | | OOM | | |
| | LEFTNet | $0.524 \pm 0.074$ | $0.508 \pm 0.038$ | $0.189 \pm 0.066$ | $0.431 \pm 0.046$ |
| Hierarchical | SchNet | $0.476 \pm 0.017$ | $0.523 \pm 0.014$ | $0.072 \pm 0.021$ | $0.424 \pm 0.016$ |
| | DimeNet++ | | OOM | | |
| | EGNN | $0.387 \pm 0.023$ | $0.461 \pm 0.020$ | $0.078 \pm 0.043$ | $0.390 \pm 0.016$ |
| | ET | $0.547 \pm 0.045$ | $0.516 \pm 0.019$ | $0.100 \pm 0.111$ | $0.438 \pm 0.029$ |
| | GemNet | | OOM | | |
| | MACE | $0.580 \pm 0.075$ | $0.476 \pm 0.048$ | $0.282 \pm 0.036$ | $0.470 \pm 0.016$ |
| | Equiformer | | OOM | | |
| | LEFTNet | $0.476 \pm 0.082$ | $0.494 \pm 0.037$ | $0.151 \pm 0.019$ | $0.446 \pm 0.029$ |
| Unified | GET (ours) | $\mathbf{0.622 \pm 0.030}$ | $\mathbf{0.533 \pm 0.014}$ | $\mathbf{0.363 \pm 0.017}$ | $\mathbf{0.533 \pm 0.011}$ |

## J DETAILED RESULTS OF UNIVERSAL LEARNING OF MOLECULAR INTERACTION AFFINITY

We provide the mean and the standard deviation of three parallel experiments on the universal learning of molecular interaction affinity (§ 4.3) in Tables 11 and 12.

Table 11: The mean and the standard deviation of three runs on protein-protein affinity prediction. Methods with the suffix "-mix" are trained on the mixed dataset of protein-protein affinity and ligand binding affinity. The best results are marked in bold and the second best are underlined.

| Repr. | Model | Rigid | Medium | Flexible | All |
|-------|-------|-------|--------|----------|-----|
| | | Pearson↑ | | | |
| Block | SchNet | $0.542 \pm 0.012$ | $0.504 \pm 0.020$ | $0.102 \pm 0.019$ | $0.439 \pm 0.016$ |
| | SchNet-mix | $0.553 \pm 0.029$ | $0.507 \pm 0.011$ | $0.093 \pm 0.041$ | $0.434 \pm 0.011$ |
| | ET | $0.575 \pm 0.041$ | $0.470 \pm 0.024$ | $0.087 \pm 0.024$ | $0.424 \pm 0.021$ |
| | ET-mix | $0.579 \pm 0.028$ | $0.502 \pm 0.019$ | $0.179 \pm 0.044$ | $0.457 \pm 0.011$ |
| | MACE | $0.621 \pm 0.022$ | $0.450 \pm 0.027$ | $0.307 \pm 0.041$ | $0.470 \pm 0.015$ |
| | MACE-mix | $0.572 \pm 0.135$ | $0.353 \pm 0.040$ | $0.170 \pm 0.046$ | $0.372 \pm 0.042$ |
| | LEFTNet | $0.563 \pm 0.035$ | $0.497 \pm 0.018$ | $0.202 \pm 0.016$ | $0.452 \pm 0.013$ |
| | LEFTNet-mix | $0.522 \pm 0.042$ | $0.544 \pm 0.009$ | $0.152 \pm 0.058$ | $0.450 \pm 0.008$ |
| Atom | SchNet | $0.592 \pm 0.007$ | $0.522 \pm 0.010$ | $-0.038 \pm 0.016$ | $0.369 \pm 0.007$ |
| | SchNet-mix | $0.625 \pm 0.017$ | $0.520 \pm 0.021$ | $-0.012 \pm 0.049$ | $0.421 \pm 0.019$ |
| | ET | $0.609 \pm 0.023$ | $0.486 \pm 0.004$ | $0.049 \pm 0.009$ | $0.401 \pm 0.005$ |
| | ET-mix | $0.618 \pm 0.048$ | $0.444 \pm 0.027$ | $0.057 \pm 0.125$ | $0.382 \pm 0.029$ |
| | MACE | $0.653 \pm 0.066$ | $0.499 \pm 0.053$ | $0.241 \pm 0.061$ | $0.463 \pm 0.052$ |
| | MACE-mix | $0.579 \pm 0.009$ | $0.484 \pm 0.056$ | $0.197 \pm 0.021$ | $0.444 \pm 0.024$ |
| | LEFTNet | $0.583 \pm 0.080$ | $0.510 \pm 0.029$ | $0.243 \pm 0.091$ | $0.448 \pm 0.046$ |
| | LEFTNet-mix | $\underline{0.688 \pm 0.021}$ | $0.532 \pm 0.021$ | $0.244 \pm 0.061$ | $0.476 \pm 0.023$ |
| Hierarchical | SchNet | $0.542 \pm 0.028$ | $0.507 \pm 0.020$ | $0.098 \pm 0.011$ | $0.438 \pm 0.017$ |
| | SchNet-mix | $0.524 \pm 0.031$ | $0.515 \pm 0.011$ | $0.135 \pm 0.077$ | $0.429 \pm 0.025$ |
| | ET | $0.572 \pm 0.051$ | $0.498 \pm 0.025$ | $0.101 \pm 0.093$ | $0.438 \pm 0.026$ |
| | ET-mix | $0.494 \pm 0.100$ | $0.501 \pm 0.007$ | $0.130 \pm 0.055$ | $0.412 \pm 0.035$ |
| | MACE | $0.616 \pm 0.069$ | $0.461 \pm 0.050$ | $0.275 \pm 0.032$ | $0.466 \pm 0.020$ |
| | MACE-mix | $0.525 \pm 0.122$ | $0.336 \pm 0.067$ | $0.060 \pm 0.114$ | $0.324 \pm 0.076$ |
| | LEFTNet | $0.533 \pm 0.059$ | $0.494 \pm 0.026$ | $0.165 \pm 0.031$ | $0.445 \pm 0.024$ |
| | LEFTNet-mix | $0.594 \pm 0.059$ | $\mathbf{0.543 \pm 0.016}$ | $0.166 \pm 0.109$ | $0.472 \pm 0.020$ |
| Unified | GET (ours) | $0.670 \pm 0.017$ | $0.512 \pm 0.010$ | $\underline{0.381 \pm 0.014}$ | $\underline{0.514 \pm 0.011}$ |
| | GET-mix (ours) | $\mathbf{0.697 \pm 0.003}$ | $\underline{0.533 \pm 0.004}$ | $\mathbf{0.389 \pm 0.009}$ | $\mathbf{0.519 \pm 0.004}$ |
| | | Spearman↑ | | | |
| Block | SchNet | $0.476 \pm 0.015$ | $0.520 \pm 0.013$ | $0.068 \pm 0.009$ | $0.427 \pm 0.012$ |
| | SchNet-mix | $0.497 \pm 0.044$ | $0.527 \pm 0.009$ | $0.042 \pm 0.031$ | $0.426 \pm 0.007$ |
| | ET | $0.552 \pm 0.039$ | $0.482 \pm 0.025$ | $0.090 \pm 0.062$ | $0.415 \pm 0.027$ |
| | ET-mix | $0.550 \pm 0.039$ | $0.524 \pm 0.019$ | $0.188 \pm 0.070$ | $0.472 \pm 0.019$ |
| | MACE | $0.596 \pm 0.047$ | $0.450 \pm 0.014$ | $0.306 \pm 0.029$ | $0.466 \pm 0.011$ |
| | MACE-mix | $0.526 \pm 0.129$ | $0.366 \pm 0.023$ | $0.193 \pm 0.030$ | $0.370 \pm 0.030$ |
| | LEFTNet | $0.515 \pm 0.039$ | $0.492 \pm 0.020$ | $0.193 \pm 0.023$ | $0.452 \pm 0.013$ |
| | LEFTNet-mix | $0.505 \pm 0.048$ | $0.543 \pm 0.028$ | $0.147 \pm 0.086$ | $0.439 \pm 0.014$ |
| Atom | SchNet | $0.546 \pm 0.005$ | $0.512 \pm 0.007$ | $0.028 \pm 0.032$ | $0.404 \pm 0.016$ |
| | SchNet-mix | $0.557 \pm 0.042$ | $0.516 \pm 0.033$ | $0.036 \pm 0.010$ | $0.428 \pm 0.022$ |
| | ET | $0.582 \pm 0.025$ | $0.487 \pm 0.002$ | $0.117 \pm 0.008$ | $0.436 \pm 0.004$ |
| | ET-mix | $0.608 \pm 0.040$ | $0.453 \pm 0.037$ | $0.058 \pm 0.135$ | $0.394 \pm 0.027$ |
| | MACE | $0.619 \pm 0.037$ | $0.487 \pm 0.049$ | $0.221 \pm 0.064$ | $0.449 \pm 0.052$ |
| | MACE-mix | $0.504 \pm 0.047$ | $0.483 \pm 0.064$ | $0.226 \pm 0.046$ | $0.449 \pm 0.029$ |
| | LEFTNet | $0.524 \pm 0.074$ | $0.508 \pm 0.038$ | $0.189 \pm 0.066$ | $0.431 \pm 0.046$ |
| | LEFTNet-mix | $\mathbf{0.634 \pm 0.060}$ | $0.518 \pm 0.026$ | $0.216 \pm 0.044$ | $0.455 \pm 0.020$ |
| Hierarchical | SchNet | $0.476 \pm 0.017$ | $0.523 \pm 0.014$ | $0.072 \pm 0.021$ | $0.424 \pm 0.016$ |
| | SchNet-mix | $0.487 \pm 0.027$ | $0.532 \pm 0.007$ | $0.096 \pm 0.053$ | $0.412 \pm 0.024$ |
| | ET | $0.547 \pm 0.045$ | $0.516 \pm 0.019$ | $0.100 \pm 0.111$ | $0.438 \pm 0.029$ |
| | ET-mix | $0.446 \pm 0.116$ | $0.499 \pm 0.009$ | $0.143 \pm 0.090$ | $0.408 \pm 0.042$ |
| | MACE | $0.580 \pm 0.075$ | $0.476 \pm 0.048$ | $0.282 \pm 0.036$ | $0.470 \pm 0.016$ |
| | MACE-mix | $0.484 \pm 0.098$ | $0.340 \pm 0.061$ | $0.086 \pm 0.125$ | $0.324 \pm 0.076$ |
| | LEFTNet | $0.476 \pm 0.082$ | $0.494 \pm 0.037$ | $0.151 \pm 0.019$ | $0.446 \pm 0.029$ |
| | LEFTNet-mix | $0.572 \pm 0.072$ | $\underline{0.553 \pm 0.029}$ | $0.143 \pm 0.124$ | $0.473 \pm 0.029$ |
| Unified | GET (ours) | $0.622 \pm 0.030$ | $0.533 \pm 0.014$ | $\underline{0.363 \pm 0.017}$ | $\underline{0.533 \pm 0.011}$ |
| | GET-mix (ours) | $\underline{0.632 \pm 0.025}$ | $\mathbf{0.555 \pm 0.008}$ | $\mathbf{0.391 \pm 0.007}$ | $\mathbf{0.537 \pm 0.003}$ |

Table 12: The mean and the standard deviation of three runs on ligand binding affinity prediction. Methods with the suffix "-mix" are trained on the mixed dataset of protein-protein affinity and ligand binding affinity. The best results are marked in bold and the second best are underlined.

| Repr. | Model | LBA | | |
|---|---|---|---|---|
| | | RMSE↓ | Pearson↑ | Spearman↑ |
| Block | SchNet | $1.406 \pm 0.020$ | $0.565 \pm 0.006$ | $0.549 \pm 0.007$ |
| | SchNet-mix | $1.385 \pm 0.016$ | $0.573 \pm 0.011$ | $0.553 \pm 0.012$ |
| | ET | $1.367 \pm 0.037$ | $0.599 \pm 0.017$ | $0.584 \pm 0.025$ |
| | ET-mix | $1.423 \pm 0.054$ | $0.586 \pm 0.012$ | $0.567 \pm 0.019$ |
| | MACE | $1.385 \pm 0.006$ | $0.599 \pm 0.010$ | $0.580 \pm 0.014$ |
| | MACE-mix | $1.449 \pm 0.050$ | $0.590 \pm 0.018$ | $0.576 \pm 0.010$ |
| | LEFTNet | $1.377 \pm 0.013$ | $0.588 \pm 0.011$ | $0.576 \pm 0.010$ |
| | LEFTNet-mix | $1.433 \pm 0.016$ | $0.543 \pm 0.005$ | $0.532 \pm 0.010$ |
| Atom | SchNet | $1.357 \pm 0.017$ | $0.598 \pm 0.011$ | $0.592 \pm 0.015$ |
| | SchNet-mix | $1.365 \pm 0.010$ | $0.589 \pm 0.006$ | $0.575 \pm 0.009$ |
| | ET | $1.381 \pm 0.013$ | $0.591 \pm 0.007$ | $0.583 \pm 0.009$ |
| | ET-mix | $1.448 \pm 0.122$ | $0.566 \pm 0.061$ | $0.564 \pm 0.059$ |
| | MACE | $1.411 \pm 0.029$ | $0.579 \pm 0.009$ | $0.563 \pm 0.012$ |
| | MACE-mix | $1.420 \pm 0.037$ | $0.580 \pm 0.030$ | $0.568 \pm 0.026$ |
| | LEFTNet | $1.343 \pm 0.004$ | $0.610 \pm 0.004$ | $0.598 \pm 0.003$ |
| | LEFTNet-mix | $1.436 \pm 0.019$ | $0.579 \pm 0.014$ | $0.561 \pm 0.016$ |
| Hierarchical | SchNet | $1.370 \pm 0.028$ | $0.590 \pm 0.017$ | $0.571 \pm 0.028$ |
| | SchNet-mix | $1.403 \pm 0.010$ | $0.572 \pm 0.004$ | $0.554 \pm 0.004$ |
| | ET | $1.383 \pm 0.009$ | $0.580 \pm 0.008$ | $0.564 \pm 0.004$ |
| | ET-mix | $1.421 \pm 0.032$ | $0.569 \pm 0.017$ | $0.558 \pm 0.017$ |
| | MACE | $1.372 \pm 0.021$ | $0.612 \pm 0.010$ | $0.592 \pm 0.010$ |
| | MACE-mix | $1.432 \pm 0.019$ | $0.588 \pm 0.011$ | $0.572 \pm 0.010$ |
| | LEFTNet | $1.366 \pm 0.016$ | $0.592 \pm 0.014$ | $0.580 \pm 0.011$ |
| | LEFTNet-mix | $1.486 \pm 0.081$ | $0.556 \pm 0.001$ | $0.545 \pm 0.005$ |
| Unified | GET (ours) | $\mathbf{1.327 \pm 0.005}$ | $\underline{0.620 \pm 0.004}$ | $\underline{0.611 \pm 0.003}$ |
| | GET-mix (ours) | $\underline{1.329 \pm 0.008}$ | $\mathbf{0.622 \pm 0.006}$ | $\mathbf{0.615 \pm 0.008}$ |

## K  ATTENTION VISUALIZATION

We visualize the attention weights between blocks on the interface of protein-protein complexes and compare them with the energy contributions calculated by Rosetta (Alford et al., 2017), which uses physics-based force fields. It is observed that the hot spots of attention weights largely agree with those predicted by Rosetta. We present three examples in Figure 6. The values are normalized by the maximum value ($v_{\max}$) and the minimum value ($v_{\min}$) in each figure (*i.e.* $v' = \frac{v - v_{\min}}{v_{\max} - v_{\min}}$)

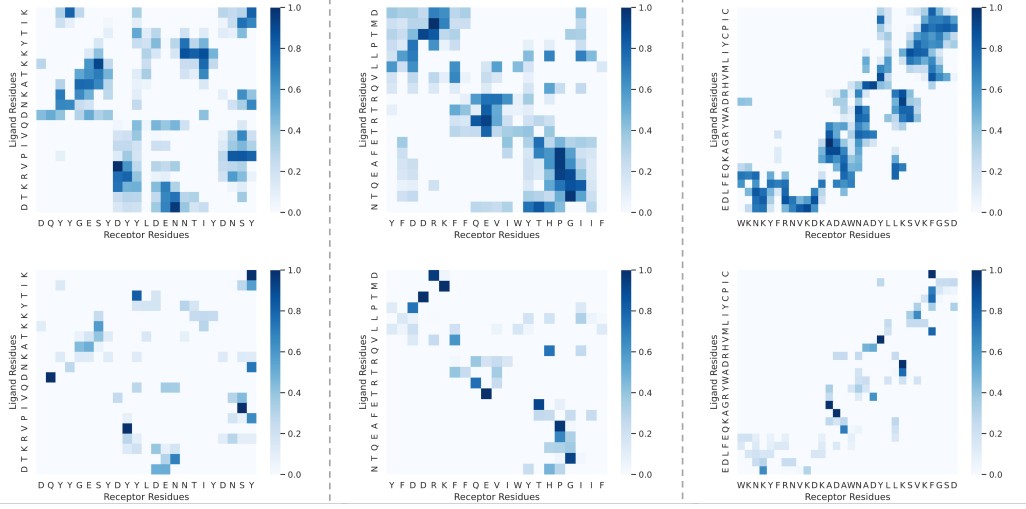

Figure 6: Attention weights of GET (upper row) and energy contributions given by Rosetta (lower row). PDB identities of the complexes are 1ahw, 1b6c, and 1gxd from left to right.

# L DISCUSSION

## L.1 LIMITATIONS

First, current evaluations mainly focus on prediction tasks in molecular interactions. Generative tasks are another major branch in learning molecular interactions (Luo et al., 2021; Liu et al., 2022; Peng et al., 2022). Designing generative algorithms for the proposed unified representation is non-trivial, and we leave this for future work. Further, it is also possible to generalize atom-level knowledge in other scenarios apart from molecular interactions (*e.g.* tasks concerning bare molecules). For instance, universal pretraining on different molecular domains, which needs careful design of the unsupervised task, hence we also leave this for future work.

## L.2 IMPLICATIONS IN PRACTICAL APPLICATIONS

Firstly, we believe that the introduction of a unified molecular representation marks a significant stride in the field of geometric molecular representation learning. The challenge of data scarcity, primarily stemming from the high costs associated with wetlab experiments, has long hindered progress in this domain. Our approach posits that, despite the limited availability of data in specific molecular domains, the underlying interaction mechanisms are shared across diverse domains. Consequently, we propose a unified model capable of accommodating data from different molecular domains, presenting a promising solution to the challenge of data scarcity. The keypoint of this strategy lies in the invention of unified molecular representations and corresponding models that exhibit robust generalization across different molecular domains. Our work serves as a first step towards this vision, demonstrating that our GET benefits from training on mixed data across different domains and exhibits exceptional zero-shot ability even on entirely unseen domains.

Secondly, in practical applications such as affinity prediction, our model offers a valuable tool for leveraging abundant data from other domains to enhance predictive performance in specific, often cutting-edge, domains that suffer from data scarcity. We illustrate this feasibility through our zero-shot experiments on RNA-ligand affinity prediction in Section 4.3. By demonstrating the adaptability of our model to a new domain without the need for specific traning data, we showcase the practical utility of our approach in scenarios where data is limited.

