# OpenReview forum: "Generalist Equivariant Transformer Towards 3D Molecular Interaction Learning"
_ICLR.cc/2024/Conference — Submitted to ICLR 2024_

### Official Review · Reviewer_KrxA · 2023-10-26

**Soundness:** 2 fair
**Presentation:** 2 fair
**Contribution:** 2 fair
**Rating:** 6
**Confidence:** 5

**Summary:**

The paper proposes a Generalist Equivariant Transformer (GET) to develop a universal representation of 3D molecular complexes and the GET model for capturing domain-specific hierarchies and domain-agnostic interaction physics.

**Strengths:**

* The paper is well-motivated and well-written.
* The proposed geometric graph of sets can be applied to various types of molecules, unifying the encoding process across different molecular structures.

**Weaknesses:**

Related concerns are discussed in the questions section.

**Questions:**

* The paper includes comparisons with several models, such as SchNet, DimeNet++, EGNN, and ET. However, there are other models that exhibit superior performance, including GemNet, NeurIPS, Allegro, MACE, etc. The paper would benefit from a more comprehensive comparison with existing methods. While the experimental results demonstrate the effectiveness of the proposed method, a more detailed comparison with other state-of-the-art models would strengthen the paper's claims. Additionally, it is important to cite these related works, as they are highly relevant to the topic of equivariant GNNs.
* Can the authors discuss the implications in practical applications?
* Typo: The model compared with GET is TorchMDNet (ET), not TorchMD.

---

> ### Author Response · Authors · 2023-11-18
> **Response to Reviewer KrxA**
>
> We sincerely thank your constructive comments. We address your concerns below and have revised our paper accordingly.
>
> > Q1: The paper includes comparisons with several models, such as SchNet, DimeNet++, EGNN, and ET. However, there are other models that exhibit superior performance, including GemNet, NeurIPS, Allegro, MACE, etc. The paper would benefit from a more comprehensive comparison with existing methods. While the experimental results demonstrate the effectiveness of the proposed method, a more detailed comparison with other state-of-the-art models would strengthen the paper's claims. Additionally, it is important to cite these related works, as they are highly relevant to the topic of equivariant GNNs.
>
> Thanks for the valuable suggestion! We appreciate your recommendation to conduct a more comprehensive comparison with existing state-of-the-art models. In response to your feedback, we have augmented our experimental comparisons by including four strong baselines, namely GemNet, Equiformer, MACE, and LEFTNet. MACE was selected due to its superior performance among the suggested baselines according to its paper [A]. Our GET consistently outperforms these additional strong baselines, reaffirming its effectiveness and superiority. We have also expanded the discussion in the related works section to provide readers with a comprehensive understanding of the landscape of existing methods and emphasize the unique contributions and strengths of our proposed approach.
>
>
> [A] Batatia, I., Kovacs, D. P., Simm, G., Ortner, C., & Csányi, G. (2022). MACE: Higher order equivariant message passing neural networks for fast and accurate force fields. Advances in Neural Information Processing Systems, 35, 11423-11436.
>
> > Q2: Can the authors discuss the implications in practical applications?
>
> Thank you for raising the question regarding the practical implications of our work.
>
> Firstly, we believe that the introduction of a unified molecular representation marks a significant stride in the field of geometric molecular representation learning. The challenge of data scarcity, primarily stemming from the high costs associated with wetlab experiments, has long hindered progress in this domain. Our approach posits that, despite the limited availability of data in specific molecular domains, the underlying interaction mechanisms are shared across diverse domains. Consequently, we propose a unified model capable of accommodating data from different molecular domains, presenting a promising solution to the challenge of data scarcity. The keypoint of this strategy lies in the invention of unified molecular representations and corresponding models that exhibit robust generalization across different molecular domains. Our work serves as a first step towards this vision, demonstrating that our GET benefits from training on mixed data across different domains and exhibits exceptional zero-shot ability even on entirely unseen domains.
>
> Secondly, in practical applications such as affinity prediction, our model offers a valuable tool for leveraging abundant data from other domains to enhance predictive performance in specific domains that suffer from data scarcity. We illustrate this feasibility through our zero-shot experiments on RNA-ligand affinity prediction in Section 4.3. By demonstrating the adaptability of our model to a new domain without the need for specific traning data, we showcase the practical utility of our approach in scenarios where data is limited.
>
> We have revised the manuscript (Appendix L) to further emphasize these points, and we hope this clarifies the practical significance of our proposed method.
>
> > Q3: Typo: The model compared with GET is TorchMDNet (ET), not TorchMD.
>
> We are sorry for our choice of ambiguous abbreviation. We have corrected it in the revision.

---

> ### Author Response · Authors · 2023-11-21
>
> Thank you for your valuable comments! To address your concerns, we have expanded our experiments with the results for four state-of-the-art models and discussed the practical implications. Please feel free to check if there is still any confusion. As the end of the discussion phase is approaching, we really hope we can hear from your feedback.

---

> ### Author Response · Authors · 2023-11-23
> **Looking forward to your feedback**
>
> Dear reviewer KrxA:
>
> As the discussion period draws to a close today, we are reaching out to kindly inquire whether our responses have effectively addressed any concerns you may have had. If there are any lingering concerns or if you have additional questions, we would be grateful for the opportunity to address them. Your thoughtful consideration is crucial to the success of our work and we look forward to hearing from you.

---

### Official Review · Reviewer_cg1F · 2023-10-29

**Soundness:** 3 good
**Presentation:** 2 fair
**Contribution:** 3 good
**Rating:** 5
**Confidence:** 4

**Summary:**

The paper proposes a unified representation of molecules as geometric graphs of sets, by performing all-atom analysis via hierarchical processing.
The model is based on the Transformer architecture where atom-level cross-attention is performed between the atoms of the same block (a subset of atoms) and where another attention is then applied at the block level. Equivariant layer norm and feed-forward layers are applied to maintain the equivariance of the coordinates.
Several experiments on molecular interactions show the superiority of the method.

**Strengths:**

1. The paper is original in its hierarchical approach and equivariant design.
2. The overall performances are significant

**Weaknesses:**

1. The paper is not clear: what makes the contributions' novelty hard to understand. Please see Question 1.
2. The assessment is not informative. Please see Question 2.

**Questions:**

The paper needs refinement in order to be clearer by providing motivations and explanations and most importantly better emphasize the technical contributions of the work.

1. Clarity:

Equation (3) is not well defined (the MLP/RBF used) it seems closely related to Shnet or Physnet [3]. Also, learning/refining the self-attention has been investigated (e.g. [1]).

Unclear how (5) is different from (Jin and al.)'s pooling.

There is some lack of motivation as to why one needs to keep track of the coordinates (which requires all the equivariant design of the non-self-attention layers) rather than working with equivariant measures such as the interatomic pairwise distances as most works do (and using other metrics too, e.g. C-RMSD [3,4]).

It is unclear (at least at first sight without looking at the appendix) how equation (10) is E(3)-equivariant (especially rotation).

Most importantly (and that's related to question 2), it's unclear (given the well-known properties of Transformers with long-range dependencies processing) why hierarchical processing is better than atom-level analysis.

2. Assessment:

In order to have a good comparison with the baselines, one should add the models' complexity and capacity comparison.
Thus, while comparing to old baselines such as Shnet or Dimenet++ (Tables 2 and 3), one should compare the difference in capacity and complexity of the models which still perform well and should be much more efficient.

[1]  Geometric transformer for end-to-end molecule properties prediction.

[2]  PhysNet: A Neural Network for Predicting Energies, Forces, Dipole Moments, and Partial Charges

[3] Molecular geometry prediction using a deep generative graph neural network
[4] Diffusion-based molecule generation with informative prior bridges

---

> ### Author Response · Authors · 2023-11-18
> **Response to Reviewer cg1F (1/3)**
>
> We sincerely thank your constructive comments. We address your concerns below and have revised our paper accordingly.
>
> > Q1: Equation (3) is not well defined (the MLP/RBF used) it seems closely related to Shnet or Physnet [3].
>
> We apologize for the unclear presentation. The MLP is a 2-layer multi-layer perceptron with SiLU [A] activation. The RBF function embeds distances into $d_{\text{rbf}}$-dimensional vectors with the following definition:
>
> $$\text{RBF}(d)[k] = u(\frac{d}{c})* \exp(-\frac{|d-\mu_k|}{2 \gamma}), 1\leq k \leq d_{\text{rbf}}$$
>
> where $\mu_k$ is uniformly distributed in $[0, c]$($c=7.0$ in our paper), $\gamma = \mu_k - \mu_{k-1} = \frac{c}{d_{\text{rbf}}}$, and $u(x)$ is the polynomial envelope function:
>
> $$u(x) = 1 - \frac{(p+1)(p+2)}{2} x^p + p(p+2)x^{p+1} - \frac{p(p+1)}{2}x^{p+2}$$
>
> We set $p=5$ in our paper. The reason why we apply the RBF function is inspired by [A][B] for its promising performance. We have added the definition to the revision.
>
> [A] Schütt, K., Kindermans, P. J., Sauceda Felix, H. E., Chmiela, S., Tkatchenko, A., & Müller, K. R. (2017). Schnet: A continuous-filter convolutional neural network for modeling quantum interactions. Advances in neural information processing systems, 30.
>
> [B] Gasteiger, J., Groß, J., & Günnemann, S. (2020). Directional message passing for molecular graphs. arXiv preprint arXiv:2003.03123.
>
> > Q2: Also, learning/refining the self-attention has been investigated (e.g. [1]).
> >
> > [1] Geometric transformer for end-to-end molecule properties prediction.
>
> Thanks for the comment. We would like to point out that our GET is directly implemented on 3D coordinates with equivariance, while [1] actually uses pairwise distance matrix to represent the geometry to achieve invariance. Moreover, we inject the inductive bias of custom hierarchy into message passing, while [1] is implemented on atomic level only. We have updated the related work with more discussion on the difference between our work and previous literature on geometric transformers/GNNs.
>
> > Q3: Unclear how (5) is different from (Jin and al.)'s pooling.
>
> We are sorry we did not express the difference clear enough. Equation (5) is the pooling on atomic pairwise relations to obtain the relation between two blocks, while (Jin and al.)'s pooling is implemented on atoms within each block to obtain the block-level representations. In particular, (Jin and al.)'s method first pools atom-level features into residue-level features, and then conducts message passing on the residue-level graphs. In constrast, our bilevel method retains the information of both atom-level and residue-level features simultaneously in each layer, namely, allowing both attention-based message passing between the atoms winthin the same residue and the neighbor residues.

---

> ### Author Response · Authors · 2023-11-18
> **Response to Reviewer cg1F (2/3)**
>
> > Q4: There is some lack of motivation as to why one needs to keep track of the coordinates (which requires all the equivariant design of the non-self-attention layers) rather than working with equivariant measures such as the interatomic pairwise distances as most works do (and using other metrics too, e.g. C-RMSD [3,4]).
> >
> > [3] Molecular geometry prediction using a deep generative graph neural network
> >
> > [4] Diffusion-based molecule generation with informative prior bridges
>
> Thank you for raising the insightful question regarding equivariant designs. We would like to answer this question from two aspects: theoretical expressivity and practical efficiency.
>
> **Theoretical Expressivity:**
>
> Existing theoretical work [A] (Theorem 8) has established that equivariant methods, which involve keeping track of coordinates, offer greater expressivity compared to invariant methods relying on pairwise distances (please see Theorem 8 in [A]). Specifically, the study reveals that equivariant Graph Neural Networks (GNNs) share expressivity with the Geometric Weisfeiler-Leman Test (GWL), while invariant counterparts align with the Invariant GWL (IGWL). The crucial distinction emerges in scenarios commonly encountered in molecular representation learning, where detecting isomorphic geometric graphs pose a challenge for invariant methods due to the lack of directional geometry in pairwise distances. In contrast, equivariant counterparts, such as our proposed GET, effectively handle these cases, enhancing the model's capacity to discern intricate geometric relationships within molecular structures.
>
> **Practical Efficiency:**
>
> Furthermore, the practical efficiency of our approach is necessary for addressing the challenges associated with scalability in molecular interaction graphs. Fully connected graphs, often implemented in invariant methods, exhibit $O(n^2)$ complexity ($n$ being the number of nodes), making them less viable for large graphs. Most existing methods relying on pairwise distances use $n \times n$ distance matrices, limiting their application to small molecular graphs with fewer than 50 nodes [B][C]. In our experiments, molecular interaction graphs are considerably larger, ranging from 230 to 872 atom nodes on average, as they incorporate extensive biomolecular structures. The equivariant nature of our GET, with linear complexity relative to the number of nodes, ensures its scalability to these larger graphs in practical settings.
>
> We hope this clarification provides a thorough understanding of our motivation for choosing equivariant designs in GET.
>
> [A] Joshi, C.K., Bodnar, C., Mathis, S.V., Cohen, T. &amp; Lio, P.. (2023). On the Expressive Power of Geometric Graph Neural Networks. Proceedings of the 40th International Conference on Machine Learning, in Proceedings of Machine Learning Research 202:15330-15355 Available from https://proceedings.mlr.press/v202/joshi23a.html.
>
> [B] Molecular geometry prediction using a deep generative graph neural network
>
> [C] Diffusion-based molecule generation with informative prior bridges
>
> > Q5: It is unclear (at least at first sight without looking at the appendix) how equation (10) is E(3)-equivariant (especially rotation).
>
> We apologize for the mistake in the presentation of equation (10). In equation (10), $\sigma_m$ should be a one-dimension scalar used to scale the geometric message vector $\vec{\mathbf{m}}_{ij, p}\in \mathbb{R}^{3}$. Intuitively, this operation only changes the length of the vector but maintains the direction, thus it is equivariant towards $O(3)$ transformations. We have corrected the equation in the revision.

---

> ### Author Response · Authors · 2023-11-18
> **Response to Reviewer cg1F (3/3)**
>
> > Q6: Most importantly (and that's related to question 2), it's unclear (given the well-known properties of Transformers with long-range dependencies processing) why hierarchical processing is better than atom-level analysis.
>
> Thank you for your thoughtful consideration and raising this important question.
>
> First, high complexity ($n^2$) is needed to achieve long-range dependencies processing for normal Transformers, which might not be applicable here. As mentioned in the previous answer to Q4, molecular interaction graphs can be extremely large, incorporating a multitude of atoms and interactions. In the context of fully connected attention matrices in Transformers, the computational complexity becomes a significant concern with such large atom-level graph sizes.
>
> Furthermore, the choice of hierarchical processing is informed by the need to capture and preserve block-level geometry, which plays a crucial role in certain molecular interaction tasks (e.g. PPA). Atom-level analysis, while valuable in capturing local interactions, may overlook the broader spatial arrangements and relationships between molecular components at higher levels of abstraction. Our bilevel representation allows us to retain and leverage the block-level and atom-level geometry simultaneously, enhancing the model's capacity to discern complex spatial patterns and dependencies within and between the molecular structures.
>
> We appreciate your concern and hope this explanation sheds light on our rationale for choosing hierarchical processing.
>
> > Q7: In order to have a good comparison with the baselines, one should add the models' complexity and capacity comparison. Thus, while comparing to old baselines such as Shnet or Dimenet++ (Tables 2 and 3), one should compare the difference in capacity and complexity of the models which still perform well and should be much more efficient.
>
> Thank you for the valuable suggestion!
>
> In our revised manuscript, we have thoughtfully addressed the comparison of model complexity and capacity. Detailed information on the number of parameters and training times for all baselines, both old and new, is available in Table 9 in Appendix G.
>
> When comparing GET to simpler yet weaker baselines (e.g., SchNet and EGNN), it is evident that GET may have more parameters and a slower training speed. However, it's crucial to note that GET exhibits competitive parameter and computation efficiency when compared with more complex yet stronger baselines, such as Equiformer, MACE, and LEFTNet. It's worth mentioning that a significant portion of parameters in GET is attributed to the Feedforward Neural Network (FFN) that projects latent features to higher dimensions in intermediate layers, aligning with the structure of vanilla Transformers. Without this part, GET has the least parameters among all the models.
>
> Moreover, the throughput of GET is comparable to both atom-level and hierarchical counterparts. This aligns with the model's design, which considers both block-level and atom-level geometry. To further elucidate the efficiency of our approach, we provide a complexity analysis of GET in Appendix D, showing its linear complexity concerning the number of nodes in the graph. This characteristic indicates favorable scalability to large graphs in practical settings.
>
> In conclusion, our model maintains a moderate number of parameters and efficient training speed while consistently outperforming baselines across multiple evaluation experiments. We believe these detailed analyses contribute to a comprehensive understanding of the efficiency and effectiveness of our proposed model.

---

> ### Author Response · Authors · 2023-11-21
>
> Thank you for the valuable suggestions! To alleviate your concerns, we have clarified the motivation for our equivariant and bilevel design, as well as analyzed the capacity and complexity of different models. Please feel free to check if there is still any problem. As the end of the discussion phase is drawing near, we really hope we can hear from your feedback.

---

> > ### Comment · Reviewer_cg1F · 2023-11-21
> >
> > Thank you for your answers and rebuttal.
> >
> > Regarding Q6:
> >
> > My concern was the following: given the huge context (now reaching millions) modern LLMs can handle, why atom-level analysis is worse than the hierarchical one? It can be practically performed (obviously with more or less strong resources) so complexity is not an issue, and should theoretically encode all levels of representation for optimal prediction.
> >
> > Regarding Q7:
> >
> > Other close baselines (Equiformer) seem to have close performance with much fewer parameters. How does it support the gain in accuracy claim?

---

> > > ### Author Response · Authors · 2023-11-22
> > > **Further responses**
> > >
> > > Thanks for your valuable feedback! We provide responses to your further concerns as follows.
> > >
> > > > My concern was the following: given the huge context (now reaching millions) modern LLMs can handle, why atom-level analysis is worse than the hierarchical one? It can be practically performed (obviously with more or less strong resources) so complexity is not an issue, and should theoretically encode all levels of representation for optimal prediction.
> > >
> > > Thank you for raising a crucial point regarding the preference for hierarchical analysis over atom-level analysis in the context of large language models (LLMs). We fully understand your concern about the potential benefits of atom-level analysis given the vast context that modern LLMs can handle. Here, we provide a comprehensive response to your quesion:
> > >
> > > **Data Scale and Efficiency**:
> > >
> > > It's acknowledged that LLMs achieve their remarkable abilities through a massive data scale, ranging from billions ($10^9$) to trillions ($10^{12}$) of data points [A]. While text data is easily obtainable for LLMs, the availability of molecular structures is limited due to the high cost of wetlab experiments. Over the decades (1970s to 2020s), only 200k ($10^5$) protein structures have been resolved, among which only 60k are unique structures, and the annual growth rate is around 10%. Extrapolating this growth suggests a considerable timeframe to reach a trillion structures. This stark contrast in data scale emphasizes the importance of data efficiency.
> > >
> > > In light of the limitations in data availability, innovation to make the most out of existing data becomes crucial. Delving into hierarchical approaches allows us to explore more efficient ways of utilizing the available data and adapting to the unique challenges posed by molecular structures. As illustrated in our empirical experiments (e.g. mix-domain data augmentation and zero-shot ability in section 4.3), our proposed bilevel representations along with GET make it effective to aggregate data from different domains, maximizing the exploitation of the existing data.
> > >
> > > **Challenges with Atom-Level Analysis**:
> > >
> > > Atom-level analysis can be hard to capture high-level geometry when applied to large molecular systems, particularly in protein-protein affinity (PPA) contexts. As shown in Table 2 (Section 4.2), hierarchical and block-level representations frequently outperform atom-level counterparts on PPA with the same backbone model. And our bilevel representation with GET consistently surpasses all the baselines, effectively capturing both atom-level and block-level geometries.
> > >
> > > Additionally, it's still crucial to be aware the limitations of LLMs. These models often decompose large contexts into windows due to quadratic complexity to sequence length [B] (so indeed the complexity is still an issue), but the 3D nature of molecular structures introduces challenges. Using windows on the sequence may split spatially close blocks into different windows, which is fatal for capturing their interactions. Further, longer contexts can lead to a significant decline in LLM performance [B]. This limitation further emphasizes the need for innovative and tailored approaches in the realm of geometric molecular representation.
> > >
> > > In conclusion, our choice of bilevel representation over atom-level analysis is driven by the constraints in data availability, the pursuit of data efficiency, and the challenges posed by the 3D nature of molecular structures. This approach aims to extract meaningful insights from existing data and adapt to the unique characteristics of molecular systems. We appreciate your insightful question and hope this response provides clarity on our methodology and its rationale.
> > >
> > > [A] Touvron, Hugo, et al. "Llama: Open and efficient foundation language models." arXiv preprint arXiv:2302.13971 (2023).
> > >
> > > [B] Liu, Nelson F., et al. "Lost in the middle: How language models use long contexts." arXiv preprint arXiv:2307.03172 (2023).
> > >
> > > > Other close baselines (Equiformer) seem to have close performance with much fewer parameters. How does it support the gain in accuracy claim?
> > >
> > > Equiformer, while demonstrating competitive performance with fewer parameters, has inherent limitations. Specifically, Equiformer exhibits instability in training and becomes memory-intensive when extended in width or depth. This limitation not only hinders its scalability but also results in extremely slow training speeds as shown in Table 9.
> > >
> > > Moreover, we would like to highlight that the reported performance metrics, such as the Spearman correlation coefficient, indicate a non-negligible improvement with GET. The observed gain from 0.496 to 0.533 in PPA and 0.591 to 0.642 in LBA reflects a substantive enhancement in predictive accuracy. We believe that such improvements are meaningful and impactful for the intended application in learning molecular interactions.
> > >
> > > We appreciate your diligence in reviewing our work and are open to further discussions or clarifications.

---

> > > ### Author Response · Authors · 2023-11-23
> > > **Looking forward to your further feedback**
> > >
> > > Dear reviewer cg1F:
> > >
> > > We would like to thank you for your engagement in the discussion. Your thoughtful feedback is important to the improvement of our work. We have provided further responses to better address your remaining concerns. As the discussion phase concludes today, we are looking forward to any further comments or feedback you may have. We hope all our efforts made above are able to resolve your concerns and will appreciate it very much if you consider raising your score if possible. Thank you once again for your time and consideration.

---

> > > > ### Comment · Reviewer_cg1F · 2023-11-23
> > > >
> > > > Thank you for your answers.
> > > >
> > > > Unfortunately, my concerns remain unmoved. The fact existing molecular datasets are of very moderate size is a good motivation for atom-level analysis in regard to the processing and analysis capabilities of modern Transformers. I was not referring to the number of data samples but to the context/molecule size. Given that modern Transformers can *efficiently process* extremely large contexts without requiring hierarchical analysis why would it not be the case for molecular tasks.
> > > > Also, the fact the ablation study did not support this was the reason for this question in the first place.

---

> > > > > ### Author Response · Authors · 2023-11-23
> > > > > **Further response**
> > > > >
> > > > > Thank you for your continued engagement and feedback. We appreciate your persistence in addressing concerns. Allow us to offer further insights to address the points raised.
> > > > >
> > > > > The intrinsic nature of large biomolecules necessitates an understanding of their basic units. Even cutting-edge models like AlphaFold 2 leverage hierarchical representations of proteins to capture essential structural information. The hierarchical approach aligns with the inherent complexity and varied scales present in large biomolecular structures (e.g. Proteins, RNAs), allowing for a more precise analysis.
> > > > >
> > > > > Regarding your reference to the ablation study, we would appreciate clarification on which specific study you are referring to with the statement, "Also, the fact the ablation study did not support this was the reason for this question in the first place." Our intention is to thoroughly address any concerns, and understanding the context of the mentioned ablation study will assist us in providing a more targeted response. In Table 2, we observe that most hierarchical models demonstrate comparable or superior performance compared to their atom-level counterparts in protein-protein affinity (PPA). This suggests that the hierarchical approach remains effective in capturing the intricacies of biomolecular tasks.
> > > > >
> > > > > Further, our choice of the bilevel representation is not solely based on the performance in one single domain, but also because of the strong generalizability across different domains. One of the major claims (in Introduction) in our paper is that the proposed bilevel representation can better depict the underlying universal physics for molecular interaction. In Table 3 (zero-shot on RNA-ligand affinity), we do observe strong generalization ability of our proposed method, which surpasses the atom-level baselines by a large margin.
> > > > >
> > > > > We hope this clarification offers a more detailed perspective on our methodology and its alignment with the unique challenges presented by molecular structures. We remain open to further discussion and appreciate your ongoing dedication to the review process.

---

### Official Review · Reviewer_cTJZ · 2023-10-31

**Soundness:** 3 good
**Presentation:** 3 good
**Contribution:** 3 good
**Rating:** 8
**Confidence:** 3

**Summary:**

This paper proposes a bilevel representation that can represent small molecules and large ones (e.g. proteins) in a unified manner. The core is to cluster the complex into a set of blocks, where intra-block connections are dense while inter-block connections are sparse. Based on this representation, the authors propose a Generalist Equivariant Transformer (GET) that captures both domain-specific hierarchies and domain-agnostic interaction physics.

**Strengths:**

1. The proposed representation and GET are reasonable and novel.
2. The paper is structured well and clearly written.
3. The authors conduct extensive experiments to support their claim, and the results are good.

**Weaknesses:**

I do not see obvious weaknesses in the proposed method and the experiments. Below are some small flaws:
1. How to construct the bilevel representation should be elaborated (e.g., what is the K of the KNN graph?).
2. Typo: (In Sec. 1) "In this paper, we approache ..."

**Questions:**

1. Compared to KNN, are there better ways to construct a block?

---

> ### Author Response · Authors · 2023-11-18
> **Response to Reviewer cTJZ**
>
> We sincerely thank you for your positive comments. We address your concerns below and have revised our paper accordingly.
>
> > Q1: How to construct the bilevel representation should be elaborated (e.g., what is the K of the KNN graph?).
>
> We apologize for the missing details. Each node in the graph consists of a set of atoms in the block. Suppose there are $n_i$ atoms in block $i$, then it is assigned with a feature matrix $\mathbf{H}_i \in \mathbb{R}^{n_i \times d}$ and a coordinate matrix $\vec{\mathbf{X}}_i \in \mathbb{R}^{n_i \times 3}$, where each row corresponds to one atom. And edges are deduced from the k-nearest neighbors of each node, where distance is defined as the minimum pairwise distances between atoms in two blocks. In this paper we use $K=9$ which is enough for supporting the competitive performance of our model. Overall, the topology of the graph is defined on the block-level geometry while atom-level instances are maintained via extensions of node features from single vectors to dynamic matrices. We have updated the manuscript for a clearer presentation in Section 3.1.
>
> > Q2: typo: (In Sec. 1) "In this paper, we approache ..."
>
> Thanks for pointing this out. We have fixed the typo in the revision.
>
> > Q3: Compared to KNN, are there better ways to construct a block?
>
> Thanks for the comment. The most common ways to construct the graph are KNN graph, radius graph (adding edges by distance cutoff), and complete graph (each node connects to all other nodes). Practically we choose KNN graph because its complexity (i.e. Number of edges) is linear to the number of nodes, which shows desired scalability to larger graphs used in our paper. It is worth mentioning that achieving good balance between complexity and performance through better construction of the graph is also a valuable research question. Existing literature has been exploring combination of different constructions [A]. This might also be related to sparse attention [B] considering that the topology of graphs can be represented as adjacent matrices.
>
> [A] Zhang, Z., Xu, M., Jamasb, A., Chenthamarakshan, V., Lozano, A., Das, P., & Tang, J. (2022). Protein representation learning by geometric structure pretraining. arXiv preprint arXiv:2203.06125.
>
> [B] Tay, Y., Dehghani, M., Bahri, D., & Metzler, D. Efficient transformers: A survey. arXiv 2020. arXiv preprint arXiv:2009.06732.

---

> ### Author Response · Authors · 2023-11-21
>
> Thank you for the positive comments! We have elaborated on the construction of the blocks and the bilevel graph to ease your concerns. As the deadline of the discussion phase is approaching, we really hope we can hear from your feedback.

---

### Official Review · Reviewer_g12k · 2023-11-03

**Soundness:** 3 good
**Presentation:** 3 good
**Contribution:** 3 good
**Rating:** 5
**Confidence:** 3

**Summary:**

This paper presents a generalist equivariant Transformer architecture for learning 3D molecular interactions. The proposed GET model is composed of bilevel attention networks, feed-forward networks, and layer normalization modules, all of which are all equivariant to E(3)-equivariant transformations. The GET model is able to simultaneously learn both the atom- and block-level information.

**Strengths:**

+ The paper is clearly written. The problem is well motivated.
+ The paper proposes a universal representation for molecular complexes. This approach has the potential to streamline and unify various interaction studies across different molecular domains.
+ The GET model ensures that all modules are E(3)-equivariant and also allows capturing interactions at block and atom scales, potentially leading to richer and more informative interaction modeling.
+ The experiments across small molecules, proteins, and nucleic acids suggest that GET has strong generalization capabilities.

**Weaknesses:**

- The reliance on domain-specific knowledge for the construction of building blocks may limit the universality of the proposed GET model (e.g., each residue in the proteins is one node). This may render the comparison with hierarchical models (e.g., GVP) a bit unfair.
- The paper would be strengthened by a more thorough comparison with state-of-the-art methods, e.g., GemNet, Equiformer, EquiformerV2, and LEFTNet.
- The experiments conducted on PPA and PBA datasets may not fully demonstrate the effectiveness of the proposed model, given their relatively small scale.
- I do not see a particular design of modeling molecular interactions in the GET architecture. It appears that it is also possible to evaluate the GET model on a broader range of molecular settings, especially bare molecules.
- The equivariant transformers and attention mechanisms raise potential concerns for practical application due to their high computational complexity, as noted in Table 9.

**Questions:**

- Including the graphical illustration (Figure 4) within the main text could provide readers with a more immediate and clear understanding of the architecture and operational flow of GET.
- Adding statistics for the PDBBind benchmark within the paper would offer a more concrete assessment for readers unfamiliar with the benchmark.

---

> ### Author Response · Authors · 2023-11-18
> **Response to Reviewer g12k (1/2)**
>
> We sincerely thank your constructive comments. We address your concerns below and have revised our paper accordingly.
>
> > Q1: The reliance on domain-specific knowledge for the construction of building blocks may limit the universality of the proposed GET model (e.g., each residue in the proteins is one node). This may render the comparison with hierarchical models (e.g., GVP) a bit unfair.
>
> Thank you for the comment. There could be some misunderstandings here. We provide the explanations below.
>
> First, we would like to emphasize that our GET model can accommodate arbitrary construction of building blocks, using wheter domain-specific kowledge or purely data-driven approaches. To show this universality, we did implement GET-PS in section 4.2, Table 3, which uses a data-driven method (i.e. principal subgraphs in the PS-VAE paper [A]) to build blocks in small molecules, and it indeed achieves brilliant performance.
>
> Second, we want to clarify that one significant strength of our GET is the capability of capturing both coarse-grained (blocks) and fine-grained (atoms) geometry. In section 4.2, for fair comparison, all the block-level baselines and the hierarchical baselines use exactly the same building blocks as our GET. However, their inability to retain bilevel geometry simultaneously makes them suboptimal compared to our proposed GET.
>
> [A] Kong, X., Huang, W., Tan, Z., & Liu, Y. (2022). Molecule generation by principal subgraph mining and assembling. Advances in Neural Information Processing Systems, 35, 2550-2563.
>
> > Q2: The paper would be strengthened by a more thorough comparison with state-of-the-art methods, e.g., GemNet, Equiformer, EquiformerV2, and LEFTNet.
>
> Thank you for your insightful recommendation to conduct a more thorough comparison with state-of-the-art methods. We have expanded our experimental comparisons by incorporating four suggested baselines: GemNet, Equiformer, MACE (suggested by Reviewer KrxA), and LEFTNet in Table 2. Besides, owing to their commendable performance and high efficiency, MACE and LEFTNet are further selected for testing on the mix-domain data augmentation in Figure 3 and zero-shot generalization ability in Table 3. Delightfully, our GET consistently outperforms all these strong baselines in all cases, demonstrating its general effectiveness and superiority. We have updated the manuscript with the results of these additional experiments.
>
> > Q3: The experiments conducted on PPA and PBA datasets may not fully demonstrate the effectiveness of the proposed model, given their relatively small scale.
>
> Thanks for the comment.
>
> We admit that PPA and LBA are of small scale, but these two datasets are our suitable choice to verify the effectiveness of GET in learning unified representations between different types of molecular interactions, namely, protein and protein in PPA, and protein and small molecule in LBA.  Actually, in the domain of molecular interaction, how to address the scarcity of data is a long-term problem.  While it is arduous to collect more data due to the high cost of wetlab experiments, we have tried to enhance the robustness and reliability of the empirical evaluations by running parallel experiments with different random seeds and data splits.
>
> Besides the experiments on  PPA and LBA, we additionally evaluate the zero-shot ability of our GET on DNA/RNA-ligand binding affinity prediction in section 4.3, as this is strong evidence for the exceptional generalizability and high data efficiency of the proposed method. It indicates our model's proficiency in adapting to completely unseen data, which is particularly valuable in addressing the challenge of data scarcity.
>
> We also want to emphasize the promising strategy outlined in our paper, which involves better utilizing all available data from different domains through the proposed unified molecular representation. This strategy aligns with the broader goal of overcoming data scarcity challenges by maximizing the information gleaned from diverse datasets.

---

> ### Author Response · Authors · 2023-11-18
> **Response to Reviewer g12k (2/2)**
>
> > Q4: I do not see a particular design of modeling molecular interactions in the GET architecture. It appears that it is also possible to evaluate the GET model on a broader range of molecular settings, especially bare molecules.
>
> Thanks for the suggestion! In this paper, we mainly focus on proposing a unified molecular representation for different types of molecular complexes, as well as a corresponding model for effective representation learning. The motivation behind is that a unified molecular representation is critical for modeling interaction geometry between multiple molecules, and endows the model with inspiring generalizability. While it is possible to evaluate GET on bare molecules, it is out of the main scope of this paper and be better left for further exploration. We have discussed this point in the limitation section (now in Appendix L due to the space limit) in the revision.
>
> > Q5: The equivariant transformers and attention mechanisms raise potential concerns for practical application due to their high computational complexity, as noted in Table 9.
>
> Thanks for the comment. We appreciate your attention to this matter, and we have carefully considered your comments.
>
> We acknowledge that equivariant transformers and attention mechanisms can introduce computational complexity. However, we would like to emphasize that our approach remains highly competitive in terms of efficiency when compared to new strong baselines (GemNet, Equiformer, MACE, and LEFTNet). In practical terms, our model offers a compelling balance between performance and computational speed, as demonstrated by the results in Table 9. Also, we have provided complexity analysis in Appendix D, which shows good scalability (linear instead of quadratic) with respect to the number of atoms in the complex graph.
>
> Moreover, despite any inherent complexity, our approach remains significantly faster than traditional computational tools (such as rosetta) in real-world applications, supporting its practicality.
>
> We have updated the manuscript to reflect these clarifications and additions to Table 9. We believe that these revisions enhance the overall quality and understanding of our work.
>
> > Q6: Including the graphical illustration (Figure 4) within the main text could provide readers with a more immediate and clear understanding of the architecture and operational flow of GET.
>
> Thanks for the advice! While it might be hard to directly include figure 4 in the main text due to the space limit, we have added a reference to it at the beginning of section 3 (Method).
>
> > Q7: Adding statistics for the PDBBind benchmark within the paper would offer a more concrete assessment for readers unfamiliar with the benchmark.
>
> Thanks for the advice! We have added necessary statistics for the PDBBind benchmark in the revision (Section 4.1 and Appendix F).

---

> ### Author Response · Authors · 2023-11-21
>
> Thank you for the insightful comments! To address your concerns, we have clarified our claims, provided additional results for four state-of-the-art models, and analyzed the complexity between different models. Please feel free to check if there is still any confusion. As the deadline of the discussion phase is drawing near, we really hope we can hear from your feedback.

---

> ### Author Response · Authors · 2023-11-23
> **Looking forward to your feedback**
>
> Dear reviewer g12k:
>
> As the discussion phase is approaching its end today, we would appreciate it a lot if you could confirm whether our response has alleviated your concerns. Your insights are invaluable, and we are eager to address any remaining concerns or questions you may have. We really hope we can hear from your feedback, which plays a crucial role in enhancing the quality of the paper.

---

> > ### Comment · Reviewer_g12k · 2023-11-23
> >
> > Thank you for your hard work. I believe most of my concerns have been cleared except for Q4. Considering that a unified molecular representation is important for modeling interaction geometry between multiple molecules, and that the GET model does not particularly model interactions of molecular interactions, it would not be unreasonable to evaluate the performance of GET on molecular tasks at first.

---

> > > ### Author Response · Authors · 2023-11-23
> > > **Further response**
> > >
> > > Thank you for your feedback, and we appreciate your acknowledgment of the efforts put into addressing your concerns. Though our primary focus is on the domain of molecular interactions, we acknowledge the concern of the reviewer on evaluations of tasks on bare molecules. Regrettably, the time constraints within the discussion phase have limited our ability to complete these experiments. However, we are committed to enhancing the comprehensiveness of our study, and we will prioritize incorporating these experiments into our revision. Your insightful comments have been instrumental in refining our work, and we are dedicated to ensuring that your concerns are thoroughly addressed in the upcoming revision. We sincerely thank you for your time and consideration.

---

### Author Response · Authors · 2023-11-18
**General Response**

We extend our sincere gratitude to all the reviewers and ACs for their dedicated efforts in reviewing our paper and providing invaluable feedbacks. In response to the constructive comments, we have implemented significant updates to enhance the clarity, comprehensiveness, and overall quality of our work. The updated contents are marked with red color in the manuscript. Here is a summary of the major revisions:
1. Responding to the suggestions from Reviewer g12k and KrxA, we have expanded our experiments to include results for a diverse set of strong methods, namely GemNet, Equiformer, MACE, and LEFTNet, in Table 2 for protein-protein affinity (PPA) and ligand-binding affinity (LBA). MACE and LEFTNet are also tested on mix-domain data augmentation (Figure 3) and zero-shot RNA-ligand affinity (Table 3). Despite the strengths of these baselines, our GET consistently outperforms them by a substantial margin, reaffirming its superiority and efficacy.
2. Reviewer g12k and cg1F suggest further analysis on capacity and complexity of the models. We have updated Table 9 in Appendix G to include a comprehensive comparison of capacity and complexity with newly added state-of-the-art methods. Our GET demonstrates moderate efficiency while maintaining exceptional performance, providing a balanced perspective on its capabilities.
3. In response to the recommendations from Reviewer g12k, cg1F, and KrxA, we have expanded the related work section to offer a more thorough summarization of the existing literature. This addition provides readers with a comprehensive overview of the contextual landscape and further strengthens the relevance of our work within the broader research domain.
4. Addressing the questions from Reviewer cTJZ and cg1F, we have added more illustrations and definitions in Section 3 (method) to provide a clearer presentation of our method.
5. In response to Reviewer g12k and KrxA, we have added a detailed discussion of limitations and implications in practical settings in Appendix L. This addition provides a more thorough understanding of the practical considerations and potential applications of our proposed method.

We believe that these updates significantly improve the paper and address the concerns raised by the reviewers. We remain open to any further suggestions or discussions and appreciate the insightful feedback that has contributed to the refinement of our work.

---

### Meta-Review · Area_Chair_LFQF · 2023-12-09

**Metareview:**

This paper studies 3D molecular interaction learning using Transformer. After rebuttals, some of the technical and experimental issues are not completely resolved. Thus a reject is recommended.

**Justification For Why Not Higher Score:**

After rebuttals, some of the technical and experimental issues are not completely resolved.

**Justification For Why Not Lower Score:**

NA

---

### Decision · Program_Chairs · 2024-01-16

Reject